



**Simulated impacts of vertical distributions of black carbon**
**aerosol on meteorology and PM$_{2.5}$ concentrations in Beijing**
**during severe haze events**
Donglin Chen[1], Hong Liao[1*], Yang Yang[1], Lei Chen[1], Delong Zhao[2], Deping Ding[2]
[1]Jiangsu Key Laboratory of Atmospheric Environment Monitoring and Pollution Control, Jiangsu
Engineering Technology Research Center of Environmental Cleaning Materials, Collaborative
Innovation Center of Atmospheric Environment and Equipment Technology, School of
Environmental Science and Engineering, Nanjing University of Information Science &
Technology, Nanjing, Jiangsu, China
[2]Beijing Weather Modification Office, Beijing 100089, China
*Correspondence to: Hong Liao (hongliao@nuist.edu.cn)



**Abstract.**
Vertical profiles of black carbon (BC) play a critical role in BC-meteorology interaction
which influences $PM_{2.5}$ (particulate matter with a diameter of 2.5 μm or less)
concentrations. In this study, we used the Weather Research and Forecasting with
Chemistry model (WRF-Chem) coupled with an improved integrated process (IPR)
analysis scheme to investigate the direct radiative effect (DRE) of BC with different
vertical profiles on meteorology and $PM_{2.5}$ concentrations in Beijing during two severe
haze events (11-12 December 2016 and 16-19 December 2016). The vertical profiles
of BC in Beijing collected by King-Air350 aircraft can be classified into two types:
the first type was characterized by decreases in BC concentration with altitude, which
was the case mainly controlled by local emissions; the second type had maximum BC
concentration around 900 hPa, which was mainly affected by regional transport from
the polluted south/southwest region. Compared with measurements in Beijing, the
model overestimated BC concentrations by 87.4 % at the surface and underestimated
BC mass by 14.9 % at altitudes of 300-900 m altitude as averaged over the two pollution
events. The BC DRE with the default vertical profiles from the model heated the air
around 300 m altitude but the warming would be stronger when BC vertical profiles
were modified for each day using observed data during the two severe haze events.
Accordingly, compared to the simulation with the default vertical profiles of BC,
planetary boundary layer heights (PBLH) were reduced further by 24.7 m (6.7%) and
6.4 m (3.8%) in Beijing and simulated $PM_{2.5}$ concentrations were higher by 9.3 μg m$^{-3}$
(4.1%) and 5.5 μg m$^{-3}$ (3.0%) over central Beijing in the first and second haze events,



respectively, with modified vertical profiles. Furthermore, we quantified by sensitivity
experiments the roles of BC vertical profiles with six exponential decline functions
($C(h)=C_0 \times e^{-h/hs}$ and $hs$=0.35, 0.48, 0.53, 0.79, 0.82 and 0.96) parameterized on the basis
of the observations and the vertical profile dominated by regional transport. A larger $hs$
leads to a sharper decline of BC concentrations with altitude (less BC at the surface and
more BC in the upper atmosphere), resulting in a stronger cooling at the surface (+0.21
with $hs$ of 0.35 vs. -0.13 °C with $hs$ of 0.96) and hence larger reductions in PBLH (larger
BC-induced increases in $PM_{2.5}$). Relative to the simulation without BC DRE, the mean
$PM_{2.5}$ concentrations were increased by 5.5 μg m$^{-3}$ (3.4%) and 7.9 μg m$^{-3}$ (4.9%) with
BC DRE when $hs$ values were 0.35 and 0.96, respectively. Our results indicate that it is
very important to have accurate vertical profiles of BC in simulations of meteorology
and $PM_{2.5}$ concentrations during haze events.


## 1. Introduction

With the rapid economic development and large increases in fossil energy consumption, haze pollution has become one of the most serious challenges in China, especially in the Beijing-Tianjin-Hebei (BTH) region (Wang et al., 2015; Zhang et al., 2019). In 2014 and 2015, the numbers of extremely serious $PM_{2.5}$ (particulate matter with an aerodynamic equivalent diameter of 2.5 μm or less) pollution days (with daily mean $PM_{2.5} > 150$ μg $m^{-3}$) in Beijing reached 45 and 54, respectively (He et al., 2017). The real-time hourly average concentration of $PM_{2.5}$ in Beijing even reached 1000 μg $m^{-3}$ during the severe haze events in January 2013, far exceeding the Chinese Ambient Air Quality Grade I Standards (35 μg $m^{-3}$ for daily mean $PM_{2.5}$) (Liu et al., 2017). With the implementation of the toughest-ever clean air policy since 2013, the observed annual mean $PM_{2.5}$ concentrations averaged over 74 cities in China fell from 61.8 μg $m^{-3}$ in 2013 to 42.0 μg $m^{-3}$ in 2017 (Zhang et al., 2016; Wang et al., 2017a; Li et al., 2019; Zhang et al., 2019). However, severe haze events still occurred in Beijing during the COVID-19 lockdown period (January-February 2020) (Huang et al., 2020; Zhu et al., 2020). Therefore, understanding the mechanisms responsible for the occurrence of severe haze is important for air quality management planning.

BC, an important component of $PM_{2.5}$, is emitted mainly from the incomplete combustion of fossil fuel, biofuel, and biomass burning. BC particles can strongly absorb solar radiation in the atmosphere, which alters the Earth's radiation balance (Bond et al., 2013; Huang et al., 2015; Hu et al., 2020). In recent years, researchers have found that the radiative effect of BC significantly affects the structure of planetary



boundary layer (PBL) during severe haze pollution events (Ding et al., 2016; Huang et
al., 2018; Wang et al., 2018; Liu et al., 2019). Ding et al. (2016) illustrated by using the
Weather Research and Forecasting model coupled with Chemistry (WRF-Chem) that
BC suppressed the development of PBL by heating the air in the upper PBL and
reducing the solar radiation at the surface in Beijing in December 2013. This process
was defined as the "dome effect" of BC by Ding et al. (2016). This "dome effect" was
also found over the Indian Ocean (Wilcox et al., 2016). BC can also change the land-
sea thermal contrast and induce circulation anomalies during severe haze events (Gao
et al., 2016b; Qiu et al., 2017; Ding et al., 2019a; Chen et al., 2021). Ding et al. (2019a)
showed by using the WRF-Chem model that, during a haze event in December 2013,
the direct radiative effect (DRE) of BC enhanced advection between land and sea by
causing a cooling (-1.0 °C) in air temperature over land and a warming (+0.8 °C) in air
temperature over sea, which transported moist air from the sea to the Yangtze River
Delta region. Qiu et al. (2017) and Chen et al. (2021) also reported by using the WRF-
Chem model that the radiative effect of BC induced strong anomalous northeasterly
winds from the sea during a haze event in North China Plain (NCP) in February 2014.
BC can influence concentrations of $PM_{2.5}$ during haze events because of its impact
on PBL and other meteorological fields (Gao et al., 2016b; Wilcox et al., 2016; Miao et
al., 2017; Qiu et al., 2017; Gao et al., 2018; Wang et al., 2019; Chen et al., 2021). Gao
et al. (2016b) used the WRF-Chem model to simulate the haze event that occurred in
the NCP in January 2010 and found a maximum increase in $PM_{2.5}$ of 14.4 $\mu g\ m^{-3}$ (5.1%)
due to the DRE of BC. Qiu et al. (2017) also analyzed the impact of BC on surface-

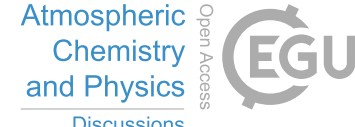

layer $PM_{2.5}$ during a haze pollution in NCP in February of 2014 by using the WRF-
Chem model and found that the average $PM_{2.5}$ concentration increased by 2.1 µg m$^{-3}$
(1.0%) owing to the DRE of BC. Chen et al. (2021) analyzed, by using the WRF-Chem
model, the DRE from the aging of BC and its impact on $PM_{2.5}$ concentration over the
BTH region during a haze event in February 2014. They found that the near-surface
$PM_{2.5}$ concentration average over BTH increased by 9.6 µg m$^{-3}$ (7.0%) due to the aging
of BC.
So far few studies examined the impacts of vertical distributions of BC aerosol on
meteorology and $PM_{2.5}$ concentrations. Wang et al. (2018) examined the role of BC at
different altitudes in influencing PBL height (PBLH) by considering a single column
using the WRF-Chem version 3.6.1. They divided the height from 150 to 2250 m evenly
into 7 layers and increased BC concentrations from 0 to 30 µg m$^{-3}$ with an increment of
2 µg m$^{-3}$ at one of the layers with the BC concentrations at the other layers fixed to 0
µg m$^{-3}$. Model results showed that the near-surface BC could increase PBLH by 0% -
4%, while BC aloft would decrease PBLH by 2% - 16% due to the warming of
atmosphere by BC. Current chemistry-climate models were reported not to be able to
represent the BC vertical profiles accurately, so sensitivity studies were carried out to
adjust vertical profiles of BC in the model by changing the vertical resolution, aerosol
microphysical scheme and emission height (Wang et al., 2019; Yang et al., 2019;
Watson-Parris et al., 2019).
In recent years, measurements of BC vertical distribution have been conducted by
aircraft during the severe haze events in Beijing, using a single particle soot photometer



(SP2) (Zhao et al., 2018; Tian et al., 2019; Zhao et al., 2019; Tian et al., 2020; Liu et
al., 2020). During the period of severe pollution from 11 to 19 December 2016, Zhao
et al. (2019) collected BC vertical profiles over Beijing by Air350 aircraft and found
that the vertical profiles can be classified into two types. The first type was
characterized by decreases in BC concentration with altitude, which was the case
mainly controlled by local emissions. The second type had maximum BC concentration
around 900 hPa, which was mainly affected by regional transport from the polluted
south/southwest region. Generally, the first type occurred more frequently than the
second type during haze events in Beijing. Observations of vertical profiles of BC in
severe haze events over Beijing in 2018 by a King-Air350 aircraft by Ding et al. (2019b)
also obtained the same types of profiles.
In this work, we use the BC vertical profiles observed during two severe haze
events (11-12 December 2016 and 16-19 December 2016) over Beijing and the online-
coupled WRF-Chem model to investigate the DRE of BC vertical profiles on
meteorology and $PM_{2.5}$ concentrations. Compared with previous studies that examined
the impact of BC on meteorology and $PM_{2.5}$, our study is the first to pay attention to the
role of BC vertical profile as well as the underlying mechanism. The description of the
model, observational datasets and numerical experiments are presented in Section 2.
Section 3 evaluates simulated meteorological and chemical variables by comparing
with observations. Section 4 compares of the DRE of BC with original and modified
vertical profiles, and Section 5 discusses the role of BC vertical profiles in influencing
meteorological parameters and $PM_{2.5}$ concentrations. The conclusions of this study are



given in Section 6.

## 2. Method

### 2.1 Model configuration

A fully coupled online Weather Research and Forecasting with Chemistry model
(WRF-Chem version 3.7.1) (Grell et al., 2005; Skamarock et al., 2008) was employed
to simulate the two severe haze events in Beijing from 7 to 20 December 2016 and the
initial 4 days are spin-up. This model adopts Lambert projection and two nested
domains with grid resolutions of 30 km (domain 01) and 10 km (domain 02). Figure 1
shows that the outer domain covers the most of China with 100 (west-east) × 100
(south-north) grid cells and the second domain covers the BTH region with 58 (west-
east) × 76 (south-north) grid cells. The number of vertical layers is 29 with the first 15
layers below 2 km for finer resolution in the PBL. Meteorological initial and boundary
conditions in this model were derived from global reanalysis data (1º×1º) of NCEP
(National Center for Environmental Prediction). MOZART-4 (Model for Ozone And
Related chemical Tracers-4) simulation results provided the initial and lateral boundary
conditions for the concentrations of chemical species in our model (Emmons et al.,

2010).

Anthropogenic emission data in year 2016 were obtained from the MEIC inventory
with a spatial resolution of 0.25º × 0.25º (Zheng et al., 2018). This inventory includes
sulfur dioxide ($SO_2$), nitrogen oxides ($NO_x$), carbon monoxide (CO), non-methane
volatile organic compounds (NMVOC), ammonia ($NH_3$), BC, organic carbon (OC),



$PM_{2.5}$, $PM_{10}$ and carbon dioxide ($CO_2$), which were categorized into agriculture,
industry, residence, transport and power generation sectors (Li et al., 2015). The
biogenic emissions were calculated online using the MEGAN (Model of Emissions of
Gases and Aerosol from Nature), including isoprene, terpene and other substances
emitted by plants (Guenther et al., 2006). Biomass burning emissions were taken from
the Fire Inventory from NCAR (FINN) datasets (Wiedinmyer et al., 2011).
The parameterization schemes of physical and chemical processes of WRF/Chem
model adopted in the study are summarized in Table 1. The Carbon-Bond Mechanism
version Z (CBMZ) is chosen to simulate the gas-phase chemistry. The aerosols scheme
is the Model for Simulating Aerosol Interactions and Chemistry (MOSAIC) which
includes sulfate, nitrate, ammonium, chloride, sodium, BC, OC and other inorganic
aerosol, and the aerosol particles are divided into 8 particle size segments. However,
the formation of secondary organic aerosol is not considered in this scheme (Zhang et
al., 2012; Gao et al., 2016a). In MOSAIC, the aerosol particles are assumed to be
internal mixture and aerosol optical properties are calculated by the volume averaging
mixing method (Barnard et al., 2010; Stelson 1990). The choice for photolysis schemes
is Fast-J photolysis scheme.
**2.2 Integrated process rate (IPR) analysis**
The IPR analysis has been widely applied to illustrate the impacts of each
physical/chemical process on the variations in $O_3$ concentrations (Zhang and Rao, 1999;
Jiang et al., 2012; Gao et al., 2017; Gao et al., 2018). The improved IPR analysis method
developed by Chen et al. (2019) in WRF-Chem model is used in this work to



quantitatively analyze the contributions of physical/chemical processes to PM$_{2.5}$
concentrations, including the contributions from the sub-grid convection (CONV),
vertical mixing (VMIX), chemistry (CHEM), regional transport (TRA), wet scavenging
(WET), emission source (EMI) and other processes (OTHER). CONV refers to the
transport within the sub-grid wet convective updrafts (Chen et al., 2019) and VMIX is
affected by atmospheric turbulence and vertical distribution of PM$_{2.5}$ concentrations
(Zhang and Rao, 1999; Gao et al., 2018). CHEM represents PM$_{2.5}$ production and loss
including gas-phase, cloud and aerosol chemistry. TRA is caused by advection, which
is highly related to wind and horizontal distribution of PM$_{2.5}$ concentrations (Gao et al.,
2018; Chen et al., 2019). WET represents the wet removal processes of aerosols. EMI
is controlled by emission source. OTHER represents the processes other than the above
6 processes in the model. The NET is the sum of all physical and chemical processes,
which matches the variations in PM$_{2.5}$ concentrations. It is worth noting that each IPR
variable is an accumulated value which is the sum of each time step.
**2.3 Observational data**

To evaluate the model performance in simulating near-surface meteorological

fields, the observed hourly temperature at 2 m (T2), relative humidity at 2 m (RH2),
wind speed at 10 m (WS10) and wind direction at 10 m (WD10) at Beijing station are
collected        from        NOAA's        National        Climatic        Data        Center
(http://gis.ncdc.noaa.gov/maps/ncei/cdo/hourly). The 3-hourly planetary boundary
layer (PBL) heights for Beijing were obtained from the Global Data Assimilation
System (GDAS) (http://ready.arl.noaa.gov/READYamet.php). The radiosonde data





(temperature and relative humidity profiles) in Beijing were obtained from the
University of Wyoming, Department of Atmospheric Science
(http://weather.uwyo.edu/upperair/sounding.html). Hourly concentrations of $PM_{2.5}$, CO,
$NO_2$, $SO_2$ and $O_3$ at Beijing Station were obtained from the China National
Environmental Monitoring Center (CNEMC, http://www.cnemc.cn/), which were used
to evaluate the model performance in simulating pollutants at the surface. Aerosol
optical depth (AOD) at 550 nm over China retrieved from MODIS (Moderate
Resolution Imaging Spectroradiometer) satellite was used to evaluate the horizontal
distribution of simulated optical properties of aerosols in this study
(https://ladsweb.modaps.eosdis.nasa.gov/). The values of daily aerosol optical depth
(AOD) at 500 nm and 675 nm in Beijing were obtained from the AERONET data set
(https://aeronet.gsfc.nasa.gov/).

The vertical profiles of BC mass concentrations in Beijing were collected by King-

Air350 aircraft using SP2 during 11-12 and 16-19 December 2016. The aircraft
departured from Shahe (~20 km to the central Beijing) (Fig. 1) at 12:00-13:00 local
time (LT) and returned during 15:00-16:00 LT, which avoided the possible diurnal
variation of the PBL among flights. Most flights could reach 2.5 km. Zhao et al. (2019)
reported that these vertical profiles of BC could be expressed as an exponential decline
function $C(h) = C_0 e^{-\frac{h}{hs}}$ except 11 December 2016, where $C(h)$ (µg m$^{-3}$) is BC
concentration at altitude $h$ (km), $C_0$ (µg m$^{-3}$) is BC concentration at the surface, and
each $hs$ value is calculated for each flight of BC vertical profile using nonlinear
regression (Table S1). Tian et al. (2019) observed a regional transport of pollution in



Beijing from 10 to 12 December 2016 using SP2 and they found a different vertical
structure of BC from that of Zhao et al. (2019), with the BC concentration at the
altitudes of 400-900 m being 1.5 times higher than the near-surface BC concentration
on 11 December 2016. More detailed information about King-Air350 aircraft dataset
can be found in Zhao et al. (2019), Ding et al. (2019b) and Tian et al. (2019).
**2.4 Numerical experiments**
To compare the DRE of BC with original and corrected vertical profiles and
quantify the role of BC vertical profiles in influencing meteorological conditions and
air pollutants, we performed the following numerical experiments as summarized in
Table 2.
1. CTRL: The control simulation with the direct and indirect radiative effects of all
aerosols (BC, OC, sulfate, nitrate, ammonium, $Na^+$, $Cl^-$ and OIN) included for the time
period of 11-20 December 2016. The vertical profiles of BC were the default ones
simulated by the model.
2. NoBCrad: The same as the CTRL simulation, except that the DRE of BC was turned
off.
3. VerBC_obs: The same as the CTRL simulation, except that the BC vertical profiles
in the model were modified according to the observed ones. The specific method will
be discussed below.
4. VerBC_hs1-6: The same as the CTRL simulation, except that the vertical profiles of
BC in the model were modified according to the exponential decline function
($C(h)=C_0\times e^{-h/hs}$). The values of $hs$ in VerBC_hs1 to VerBC_hs6 were 0.35 to 0.96 (from

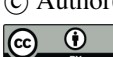



small to large), respectively.
5. VerBC_RT: The same as the VerBC_hs1-6 simulations, except that the BC vertical
profiles in the model were modified according to the observed transport BC vertical
profile on 11 December 2016 which was affected by regional transport.

In the case of NoBCrad, the BC DRE was turned off by setting the BC mass

concentration equal to zero when calculating the optical properties of BC, following the
studies of Qiu et al. (2017) and Chen et al. (2021). In VerBC_obs experiment, we
modified the simulated BC vertical profile online using the observed BC vertical profile
on the corresponding day. Firstly, we interpolated the observed BC concentrations to
the height of each layer in the model as $C_{obs\_int(i)}$. Each layer in the model has a top
height and a bottom height and we selected the middle height of this layer for
interpolation. Secondly, we used $C_{obs\_int(i)}$ to calculate the BC mass column burden in
each layer ($M_{obs\_int(i)}$) in the model, and $P_{obs\_int(i)}$ is the percentage of BC mass column
burden in each layer to the total BC mass column burden (Fig. S1) calculated by
$$M_{obs\_int(i)} = C_{obs\_int(i)} * (H_{sim\_top(i)} - H_{sim\_bot(i)}) \tag{1}$$
where $H_{sim\_top(i)}$ is the top height of layer $i$ and $H_{sim\_bot(i)}$ is the bottom height of layer $i$.
$$P_{obs\_int(i)} = \frac{M_{obs\_int(i)}}{\sum_{i=1}^{16} M_{obs\_int(i)}} * 100\% \tag{2}$$

In the VerBC_obs simulation, the simulated BC mass column burden was redistributed
to each layer below 2.5 km according to the calculated $P_{obs\_int(i)}$. These procedures
ensure that the modification of BC vertical profile for each day in the model by using
the observed data does not change the total BC mass column burden. Since the aircraft



measured only BC concentrations below 2.5 km, we modify BC profile up to the 16th
model layer (about 2.5 km in Beijing).

In the experiment of VerBC_hs1, we also used the above method to modify the BC

vertical profile by an exponential decline function which is $C(h)=C_0\times e^{-h/hs}$. However,
in cases of VerBC_hs1~6, we modified for the dates of 12 and 16-19 December. On
December 11, BC did not show an exponential decline with height due to the regional
transport. In simulation of VerBC_RT, the method and setting were the same as
VerBC_hs1~6, except that the BC vertical profile in the model was modified according
to the observed one on 11 December 2016. In VerBC_obs, VerBC_hs1~6 and
VerBC_RT cases, the steps of modifying BC vertical profiles were performed only
when the direct radiative forcing of BC was calculated. All other physical and chemical
processes still used the original BC vertical profiles simulated by the model.
**3 Model evaluation**
**3.1 Near-surface air pollutants and BC vertical profiles**

Figures 2a-2i show the horizontal distributions of simulated near-surface $PM_{2.5}$

concentrations at 2 pm LT from 11 to 19 December 2016. In BTH, high $PM_{2.5}$
concentrations of 138.4 and 90.8 μg m$^{-3}$ occurred on December 11 and 12, respectively.
The severe pollution on December 11 was caused by regional transport from the
southern heavily polluted area under a prevailing southerly air flow (Tian et al., 2019).
From December 16, $PM_{2.5}$ started to accumulate in the eastern China and the
concentrations of $PM_{2.5}$ reached highest value of 153.4 μg m$^{-3}$ on 18 December
averaged over the BTH region. The daily $PM_{2.5}$ concentrations (Fig. 2j) in Beijing had



low values during 13-15 December 2016. The severe pollution during 16-19 December
2016 was mainly affected by local emissions. We are mainly focused on the two heavy
pollution incidents (11-12 and 16-19 December 2016) in the following sections.
Results from the CTRL simulation were compared with the observed hourly
surface concentrations of $PM_{2.5}$, $NO_2$, $O_3$, CO and $SO_2$ during 11-19 December 2016 in
Beijing in Fig. 3. The observed maximum $PM_{2.5}$ concentration of 219.5 μg m$^{-3}$ occurred
on December 18, far exceeding the national air quality standard for daily $PM_{2.5}$ of 75
μg m$^{-3}$ (Wang et al., 2017a). The correlation coefficient (R), mean bias (MB),
normalized mean bias (NMB) and mean fraction bias (MFB) are summarized in Table
3. The model can reasonably reproduce the temporal variations of $PM_{2.5}$, $NO_2$, $O_3$ and
CO; the correlation coefficients between simulated and observed hourly concentrations
are 0.77, 0.78, 0.66 and 0.73, respectively. The correlation coefficient for $SO_2$ is lower
(0.38). Gao et al. (2016b) explained that WRF-Chem model cannot represent well the
$SO_2$ concentration and its change with time due to the uncertainty in $SO_2$ emissions and
missing heterogeneous oxidation. Compared with observations, the model
overestimates the concentrations of $PM_{2.5}$ and $NO_2$ in Beijing with the MBs and NMBs
of (13.2 μg m$^{-3}$, 10.0%) and (8.5 ppbv, 21.6%), respectively, and underestimates the
concentrations of $O_3$ (-0.1 ppbv, -1.2%) and CO (-0.1 ppmv, -4.9%). Overall, the model
can capture the two severe pollution events in Beijing during 11-19 December 2016.
Because of the lack of measured BC vertical profiles from 13-15 December 2016
in Beijing, Figure 4 compares only the simulated vertical profiles of BC with
observations for the two polluted events (11-12 and 16-19 December 2016). Observed



mass concentrations of BC decreased exponentially with altitude in all days except for
December 11 when regional transport of pollution dominated. On December 11, the
observed maximum mass concentration of BC (7.0 μg m$^{-3}$) occurred at 850 m altitude,
which was much higher than the surface-layer concentration of 4.7 μg m$^{-3}$. Compared
with the observed vertical profiles of BC, the model can well represent the decreases of
BC mass concentration with height on December 12 and 16-19, but cannot reproduce
the vertical profile on December 11. Averaged over the two pollution events, the
simulated BC mass concentration was overestimated by 87.4% on the ground and
underestimated by 33.1% at altitude of 1000 m compared with the observations in
Beijing. The inaccuracy of the vertical distribution of BC would lead to inaccurate
representation of the interactions between BC and PBL, especially in heavily polluted
events.
**3.2 Meteorological parameters**

On December 12-16 and 18-19, Beijing was mainly controlled by northerlies and

northwesterlies, which transported clean air mass to Beijing. On December 11 and 17,
southwesterlies brought polluted air to Beijing, as shown in Fig. 2. Nevertheless, the
average wind speed in Beijing was 5.1 m s$^{-1}$ at 850 hPa on December 17, which was
much smaller than 11.0 m s$^{-1}$ at 850 hPa on December 11, which explains that Beijing
was less affected by the regional pollution transport on December 17. Figure 5 shows
the hourly simulated and observed T2, RH2, WS10, WD10 and PBLH in Beijing from
11 to 19 December 2016. The statistical metrics are summarized in Table 3. In the two
severe haze events, the observed maximum RH2 in each day exceeded 70.0%, which

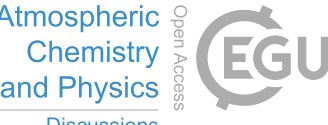

accelerated the formation of secondary aerosols (Sun et al., 2006; Wang et al., 2014).
Compared with the observations, the model can well represent the temporal variation
of T2 and RH2 with correlation coefficients of 0.77 and 0.75, respectively, but slightly
overestimates T2 with a MB of 0.1 ℃ and underestimates RH2 with a MB of 3.4%. For
WS10, observations and simulated results both show low wind speed with the mean
values of 1.5 and 1.4 m s$^{-1}$ in Beijing during the two periods of haze events. Such
meteorological condition was very beneficial to the accumulation of near-surface
pollution. The WRF-Chem model also captures the high values of WS10 from 14 to 15
December. For wind direction at 10 m, the NMB is -9.0% and the R is 0.45, which
indicates that the model can simulate the change of wind direction during the period of
heavy pollution. For PBL, the observed PBL was 118.7 m during the two severe haze
events, compared to 287.5 m during the clean period. The model can represent the
change of PBLH in Beijing from 11 to 19 December 2016 with R of 0.72. However, the
model overestimates the PBLH by 30.9 m (17.7%) in Beijing averaged over 11-19
December 2016.
The simulated and observed vertical profiles of temperature in Beijing during 11-
19 December 2016 are shown in Fig. S2. The observed temperature vertical profiles are
available only at 8:00 and 20:00. During the two severe pollution events, strong
temperature inversions below 1500 m were observed in Beijing, which inhibited
vertical mixing and caused the accumulation of pollutants near the ground. The model
captures these temperature inversions well but overestimates the inversion layer height
on December 11 and underestimates the inversion layer height from 16 to 19 December.



The inaccuracy of the simulated inversion layer height may be due to the fact that the
model cannot correctly represent the vertical profiles of BC (Fig. 4).

### 3.3 AOD and AAOD

AOD (AAOD) is the measure of aerosols (absorbing aerosols) distributed within a
column of air from the surface to the top of the atmosphere (Khor et al., 2014). Figure
S3 shows the horizontal spatial distributions of observed and simulated AOD at 550 nm
over the NCP averaged over 11-19 December 2016. The model can well simulate the
horizontal distribution of AOD, with a spatial correlation coefficient of 0.89. However,
the model underestimates AOD over the NCP region. Many previous studies have
shown that MODIS retrieval tends to overestimate AOD over NCP (Li et al., 2016; Qiu
et al., 2017). We also compared the simulated hourly AAOD at 550 nm with AERONET
AAOD in Beijing and Xianghe station in Fig. 6. The correlation coefficient between
simulations and observations is 0.85. Compared with AERONET AAOD, simulated
AAOD values at Beijing and Xianghe are overestimated by 0.02 (33.3%) and 0.02
(39.9%), respectively.

### 4. A comparison of BC DRE with original and modified vertical profiles

As shown in Fig. 4, the model does not represent well the vertical distribution of
BC concentrations during the two heavily polluted events, especially on December 11.
So, in this section, we examine the differences in the BC DRE on meteorology and
concentrations of pollutants with the original and modified vertical profiles.

### 4.1 Direct radiative effect of BC on meteorology



Figure 7 shows the atmospheric temperature and PBLH simulated from the CTRL
simulation and their changes caused by BC DRE with original profiles (CTRL minus
NoBCrad) and modified profiles (VerBC_obs minus NoBCrad), over Beijing from 11
to 19 December 2016. Light-absorbing BC heated the air at around 300 m on December
11 and 16-19, regardless of the original or modified BC vertical profiles (Fig. 7b and
7c). With the original and modified BC profiles, the maximum warming effects in the
PBL were 0.8 °C and 0.9 °C, respectively, at 14:00 LT on December 18. Although BC
concentration was the highest at the surface, the largest increase in temperature
occurred in the upper layers because of the stronger shortwave absorption efficiency of
BC at higher altitude (Ding et al., 2016; Wang et al., 2018). The warming at around 300
m resulted in a more stable stratification, thereby weakening convective motions (Gao
et al., 2018). The largest reductions in PBLH were 133.8 m (28.0%) at 14:00 LT on
December 12 and 141.2 m (59.0%) at 14:00 LT on December 18 in Beijing with original
and modified BC vertical profiles, respectively. On December 11 when regional
transport of pollution dominated, relative to the simulation with original BC profile,
simulated air temperature with modified profile was lower by about 0.5 °C within the
PBL (Fig. 7d), which was caused by the observed maximum mass concentration of BC
around 850 m altitude (Fig. 4a). Correspondingly, the maximum reduction in PBLH of
74.2 m was also simulated on December 11. On December 16-19 when local emissions
dominated, compared the effects of original BC profiles, the air temperatures at around
300 m were all higher with modified BC. The largest difference of +0.1 °C was
simulated in the PBL on December 18 (Fig. 7d).





The spatial distribution of 10-m winds averaged over the two severe haze events is
shown in Fig. S4. When the BC DRE was not considered in NoBCrad simulation, the
overall wind speed in Beijing was weak with a mean value of 3.6 m s$^{-1}$, which made it
difficult for local emissions to diffuse. The westerlies in the central part of NCP brought
relatively clean air to Beijing. Compared to the baseline of NoBCrad, the BC DRE with
original and modified vertical profiles both enhanced the northerlies north of NCP and
weakened the wind speed in central and southern Beijing.
**4.2 Direct radiative effect of BC on PM$_{2.5}$ concentration**
By altering the meteorological conditions, BC exerts feedback onto PM$_{2.5}$
concentrations. Figure 8 shows the spatial distributions of changes in near-surface
PM$_{2.5}$ induced by BC DRE with original (CTRL minus NoBCrad, Figs. 8a1-8a2) and
modified (VerBC_obs minus NoBCrad, Figs. 8b1-8b2) vertical profiles, as well as the
differences between VerBC_obs and CTRL (VerBC_obs minus CTRL, Figs. 8c1-8c2)
over Beijing in the two haze events. Because of the differences in BC-induced changes
in air temperature, wind field, and PBLH, changes in near-surface PM$_{2.5}$ concentrations
in the northern and southern Beijing were different. In the first haze event of 11-12
December, although PBLH was reduced in the northern Beijing due to BC DRE,
enhanced northerlies brought in relatively clean air to northern Beijing, leading to
decreases in near-surface PM$_{2.5}$ concentrations with maximum values of 12.5 μg m$^{-3}$
(9.4%) and 10.6 μg m$^{-3}$ (8.0%) in this region with the original and modified BC vertical
profiles, respectively. Nevertheless, PM$_{2.5}$ concentrations increased by up to 17.8 μg m$^{-}$
$^{3}$ (6.6%) and 24.0 μg m$^{-3}$ (9.3%) in the southern Beijing due to BC effect with original



and modified vertical profiles, respectively. In the second haze event of 16-19
December, the near-surface $PM_{2.5}$ concentrations increased in most areas of Beijing
with both vertical profiles. Compared to the simulation with the original profiles, the
modified profiles of BC led to larger increases in $PM_{2.5}$ concentrations over Beijing,
and the maximum differences in $PM_{2.5}$ were simulated over central Beijing, which were
9.3 $\mu g\, m^{-3}$ (3.6%) and 5.5 $\mu g\, m^{-3}$ (3.1%) in the first and second haze events, respectively.
To explain the changes in near-surface $PM_{2.5}$ concentrations in Beijing due to BC
effects, we carried out process analysis for $PM_{2.5}$ for 12:00-18:00 of each day when the
DRE of BC is the largest (Figs. 8a3, 8b3, and 8c3). From 11 to 19 December 2016,
VMIX had dominant positive contribution to changes in $PM_{2.5}$ concentration, which
reached the maximum contributions of 32.4 $\mu g\, m^{-3}$ and 33.9 $\mu g\, m^{-3}$ on December 18
with original and modified BC vertical profiles, respectively. The vertical mixing was
strongly restrained by PBLH, therefore, the decreases in PBLH caused accumulation of
$PM_{2.5}$ in the lower layers. Meanwhile, CHEM contributed 4.8 $\mu g\, m^{-3}$ and 6.1 $\mu g\, m^{-3}$ to
$PM_{2.5}$ changes because more aerosol precursors restrained in the boundary layer led to
the formation of secondary particles. TRA was the major process that had negative
contribution to the changes in $PM_{2.5}$, which can be explained by the enhanced
northerlies in the central part of NCP due to BC effects as shown in Fig S4. Relative to
the case with original BC vertical profiles, VMIX and CHEM contributions increased
largely with modified profiles, with increases of 8.6 $\mu g\, m^{-3}$ (6.5%) and 7.7 $\mu g\, m^{-3}$
(26.8%), respectively, as averaged over the two haze events, reflecting the further
decreases in PBLH (Fig. 7d).



Figure 9 shows the vertical profiles of the contributions of physical/chemical
processes to changes in PM$_{2.5}$ over Beijing due to BC DRE with original (CTRL minus
NoBCrad; Figs. 9a1 and 9b1) and modified vertical profiles (VerBC_obs minus
NoBCrad; Figs. 9a2 and 9b2) in the two haze events. In the first haze event of 11-12
December when regional transport of pollution dominated, the NET contribution to
PM$_{2.5}$ was positive below 256 m, because of the positive contribution of VMIX was
larger than the negative contribution of TRA. However, in the upper layers (from 256
to 1555 m), the contributions of VMIX and CHEM became negative with both original
and modified vertical profiles, which can be explained by the decreases in PBLH
inhibiting the transport of low-layer pollutants to the upper layer. Compared to the
original BC vertical profiles, the modified BC vertical profiles increased PM$_{2.5}$ in the
entire vertical layers below 2080 m, in which the positive contribution between 256-
757 m was caused by TRA. These results agree with the observed high concentrations
of BC at altitudes of 600-1500 m on December 11 (Fig. 4a). In the second haze event,
the NET contribution to PM$_{2.5}$ was positive below 127 m and negative at 127-504 m.
However, the effects of BC on PM$_{2.5}$ were small above 504 m because BC
concentrations decreased rapidly with altitude.
**5. Roles of BC vertical profiles**
BC has higher light-absorbing efficiency at higher altitudes (Ding et al., 2016;
Wang et al., 2018). As described in Section 2.3, the observed vertical profiles of BC on
heavily polluted days (12 and 16-19 December) can be parameterized as exponential
decline functions using nonlinear regression ($C(h)=C_0 \times e^{-h/hs}$) with $hs$ values of 0.35,



0.48, 0.53, 0.79, 0.82 and 0.96, and the profiles affected by regional transport had high
concentrations of BC at high altitudes. We conducted seven sensitivity experiments
which applied six exponential functions and one observed transport-dominated vertical
profile, as described in Section 2.4, to examine the roles of BC vertical profiles in
influencing meteorological conditions and PM$_{2.5}$ during severe haze events. In these
sensitivity experiments, we only modify the BC vertical profiles for the dates of 12 and
16-19 December. In the function of $C(h)=C_0\times e^{-h/hs}$, a larger $hs$ leads to a sharper decline
of BC concentrations with altitude (less BC at the surface and more BC in the upper
atmosphere), as shown in Fig. 10.

**5.1 Impacts of BC vertical profiles on meteorology**

Figure 11 shows the simulated changes in atmospheric temperature induced by BC
DRE with exponential functions (VerBC_hs1-6 minus NoBCrad) and with the
transport-dominated vertical profile (VerBC_RT minus NoBCrad). BC had a significant
warming effect at altitudes of 256-421 m from 12:00 to 18:00 (Fig. 7). Generally, with
the value of $hs$ gradually increasing, the BC-induced warming in the afternoon around
300 m became smaller, which can be explained by the highest mass fraction of BC at
the altitudes of 256-421 m to total BC column burden in VerBC_hs1 case (31.7%) and
the lowest percentage in VerBC_hs6 case (21.7%) among the six sensitivity
experiments (Fig. S1). The maximum warming around 300 m was 0.42 °C in
VerBC_hs1 case and 0.19 °C in VerBC_hs6 case. It should be noted that BC led to a
significant cooling effect at the surface (below 80 m) when $hs$ values were 0.79, 0.82
and 0.96, with the changes in temperature by -0.08, -0.09 and -0.13 °C, respectively.





Because more BC mass was assigned into high altitudes (above 1000 m) with higher
*hs*, less solar radiation could reach the ground (Fig. S5). These results are consistent
with those found in previous modeling and observational studies (Cappa et al., 2012;
Ferrero et al., 2014; Ding et al., 2016; Wang et al., 2018). Meanwhile, in the case of
VerBC_RT, BC also had a cooling effect of 0.30 °C at the surface (Fig. 11g). Many
studies could hardly simulate the cooling of BC at the surface, which might be caused
by the vertical profiles of BC in the model (Wang et al., 2019).
We further use the difference in temperature between the upper PBL ($T_H$; 256-421
m) and the ground ($T_L$; 0-127 m) ($\Delta T_{BC} = T_H - T_L$) averaged over 12:00-18:00 LT of 12
and 16-19 December to quantify temperature inversion caused by BC DRE. With *hs*
values increasing from 0.35 to 0.96, $\Delta T_{BC}$ increased from 0.17 to 0.42 °C, and the $\Delta T_{BC}$
value was 0.51 °C in VerBC_RT case (Fig. 12a). As a result, the reductions in PBLH
were larger with higher *hs* (Fig. 12b). The minimum decrease in PBLH was -31.9 m (-
14.3%) with *hs* value of 0.35 and the maximum decrease was 48.9 m (22.0%) with *hs*
value of 0.96, as averaged the period of 12:00-18:00 of 12, 16-19 December. In the case
of VerBC_RT, the mean PBLH was reduced by 56.9 m (25.6%) during the period of
12:00–18:00.
**5.2 Impacts of BC vertical profiles on PM$_{2.5}$ concentration**
Figure 13a shows the changes in surface-layer PM$_{2.5}$ concentration caused by BC
DRE with six exponential functions (VerBC_hs1-6 minus NoBCrad) and the transport-
dominated vertical profile (VerBC_RT minus NoBCrad) averaged over 12 December
and 16-19 December 2016. From 0:00 to 11:00, the surface-layer PM$_{2.5}$ exhibited larger



BC-induced decreases with a higher value of $hs$. This can be explained by the negative
contribution of TRA process that increased during the period of 0:00-5:00 (Figs. 13b
and 13c) when $hs$ value changed from 0.35 to 0.96. The near-surface $PM_{2.5}$
concentration was reduced by up to 9.1 μg m$^{-3}$ (6.2%) and 12.6 μg m$^{-3}$ (8.6%) at 5:00
with $hs$ values of 0.35 and 0.96, respectively. Compared to the NoBCrad case, the
surface-layer $PM_{2.5}$ concentrations were reduced by up to 13.8 μg m$^{-3}$ (9.4%) at 5:00
due to BC DRE in VerBC_RT case. From 12:00 to 18:00, the BC-induced increases in
surface-layer $PM_{2.5}$ concentrations were larger as $hs$ values are higher; relative to
NoBCrad simulation, the mean $PM_{2.5}$ concentrations were increased by 5.5 μg m$^{-3}$
(3.4%) and 7.9 μg m$^{-3}$ (4.9%) with the $hs$ values of 0.35 and 0.96, respectively. Because
the PBL was suppressed by BC DRE from 12:00 to 15:00, the contributions of VMIX
and CHEM to near-surface $PM_{2.5}$ were positive and larger in magnitude than the
negative contribution of TRA. The NET of all processes was negative from 16:00 to
18:00 due to the continuous growth of negative contribution of TRA. The negative
contribution of TRA process from 12:00 to 18:00 can be explained by the enhanced
northerlies in the central part of BTH caused by BC DRE, which transported cleaner
air mass into Beijing (Fig S6). From 19:00 to 23:00, the surface-layer $PM_{2.5}$
concentrations were decreased by BC DRE, which can be explained by the dominant
negative contribution of TRA from 19:00 to 21:00. At 22:00, the reduction in surface-
layer $PM_{2.5}$ was 7.5 μg m$^{-3}$ (4.0%) when hs value was 0.35 and 6.6 μg m$^{-3}$ (3.5%) when
$hs$ was 0.96.
**6. Conclusions**



In this study, a fully coupled online WRF-Chem model with an improved integrated
process rate (IPR) analysis scheme is employed to investigate the direct radiative effects
(DRE) of BC vertical profiles on meteorology and PM$_{2.5}$ concentrations during two
severe haze events (11-12 December 2016 and 16-19 December 2016). Compared to
the vertical profiles of BC in Beijing collected by King- Air350 aircraft using SP2, the
default vertical profiles of BC from the WRF-Chem model can capture the decreases
of BC mass concentration with altitude on December 12 and 16-19 when local
emissions dominated, but cannot reproduce the observed maximum mass concentration
of BC around 850-m altitude on December 11 when regional transport of pollutants
dominated. Averaged over the two severe pollution events, the model overestimates BC
mass concentration by 87.4% at the surface but underestimates BC by 33.1% at 1000-
m altitude compared with the observations in Beijing.
We carried out simulations with both the default original BC vertical profiles and
the modified vertical profiles using the observations (keep the column burden of BC
from the WRF-Chem but distribute BC mass vertically according to the observed
fractions of BC in individual model layers for each day). Compared with the simulation
with original BC profiles, the warming by BC DRE around 300-m altitude was stronger
with the modified profiles. Accordingly, the BC-induced reductions in PBLH in Beijing
averaged over the two severe haze events were 43.4 m (18.4%) and 55.4 m (23.5%),
respectively, with the original and modified profiles. As a result, relative to the
simulation with the original profiles, the modified profiles of BC led to larger increases
in PM$_{2.5}$ concentrations by BC DRE. The maximum differences in PM$_{2.5}$ (VerBC_obs


minus CTRL) were simulated over central Beijing, which were 9.3 µg m$^{-3}$ (4.1%) and
5.5 µg m$^{-3}$ (3.0%) in the first and second haze events, respectively. IPR analysis is used
to explain the changes in PM$_{2.5}$ concentrations caused by BC DRE. During the two
severe haze events, VMIX and CHEM had the dominant positive contributions to the
changes in surface-layer PM$_{2.5}$ due to the reductions in PBLH, and TRA had the key
negative contribution to PM$_{2.5}$ changes.

Seven sensitivity experiments were further carried out to understand the roles of

BC vertical profiles. In six assumed exponential functions ($C(h)=C_0 \times e^{-h/hs}$) with $hs$
values of 0.35, 0.48, 0.53, 0.79, 0.82 and 0.96, a larger $hs$ leads to a sharper decline of
BC concentrations with altitude (less BC at the surface and more BC in the upper
atmosphere). In all the cases, the simulated largest warming occurred at altitudes of
256-421 m. With the value of $hs$ gradually increasing, the BC-induced warming in the
afternoon around 300-m altitude became smaller, the maximum warming was 0.42 ℃
in VerBC_hs1 case ($hs$=0.35) and the minimum warming was 0.19 ℃ in VerBC_hs6
case ($hs$=0.96). While BC led to warming of 0.21, 0.08 and 0.04 ℃ at the surface when
$hs$ values were 0.35, 0.48 and 0.53, it led to a significant cooling near the surface (below
80 m) when $hs$ values were 0.79, 0.82 and 0.96, with the changes in temperature by -
0.08, -0.09 and -0.13 ℃, respectively. Stronger temperature inversion with higher $hs$
led to larger BC-induced increases in PM$_{2.5}$; relative to NoBCrad simulation, the mean
PM$_{2.5}$ concentrations were increased by 5.5 µg m$^{-3}$ (3.4%) and 7.9 µg m$^{-3}$ (4.9%) with
the $hs$ values of 0.35 and 0.96, respectively.

Results from our study highlight the importance of accurate representation of BC



vertical profiles in models, which alter the radiation balance, BC-PBL interaction, and
hence the simulated $PM_{2.5}$ concentrations. Due to the limitation of observational data,
this study was focused on the DRE of BC vertical profiles on meteorology and $PM_{2.5}$
concentration in Beijing during severe haze events. However, the results from this study
should be generally important for understanding severe haze for urban areas.

There are channels for further improvement in near-future research. We distribute

BC mass vertically according to the observed fractions of BC in individual model layers
for each day without considering the hourly variations of BC vertical profiles due to the
lack of data. Such assumed distribution of BC based on observation may not be
consistent with the dynamic processes (winds, temperature, etc.) of the atmosphere.
Further efforts are needed to examine the roles of BC vertical profiles in coupled
chemistry-weather models.



***Data availability.***
The WRF-Chem model is available at
https://www2.mmm.ucar.edu/wrf/users/downloads.html (last access: 7 July 2020).
The observations and simulation results are available upon request from the
corresponding author (hongliao@nuist.edu.cn).

***Competing interests.***
The authors declare that they have no conflict of interest.

***Author contributions.***
DC and HL designed the study and DC wrote the paper. DC performed model
simulations and analyzed the data. DZ and DD provided the observed data. YY and LC
provided technical support.

***Acknowledgements.***
This work was supported by the National Key Research and Development Program of
China (grant no. 2019YFA0606804), the National Natural Science Foundation of China
(grant no. 42021004), and the Major Research Plan of the National Social Science
Foundation (grant no.18ZDA052).




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





**Table1.** Physical and chemical options for WRF/Chem.

| WRF/Chem Model Configuration | Description |
| --- | --- |
| Microphysics scheme | Lin microphysics scheme (Wiedinmyer et al., 2011) |
| Longwave radiation scheme | RRTMG scheme (Zhao et al., 2011) |
| Shortwave radiation scheme | RRTMG scheme (Zhao et al., 2011) |
| Gas phase chemistry scheme | CBMZ (Zaveri and Peters, 1999) |
| Aerosol module | MOSAIC (Zaveri et al., 2008) |
| Photolysis scheme | Fast-J (Wild et al., 2000) |
| Boundary layer scheme | Yonsei University Scheme(YSU) (Hong et al., 2006) |
| Pavement parameterization scheme | Noah Land Surface Model scheme |







**Table 2.** Numerical experiments. Y indicates "on", and N indicates "off".

| Simulations | BC direct radiative effect (DRE) | | |
| | DRE | BC vertical profiles for calculation of DRE | |
| | Turn on/off | Types description | Modified dates |
|---|---|---|---|
| CTRL | Y | Simulated by model | No modification |
| NoBCrad | N | Simulated by model | No modification |
| VerBC_obs | Y | Modified according to intraday observations | 11-12 and 16-19 December |
| VerBC_hs1-6 | Y | Modified according to $C(h)=C_0 \times e^{-h/hs}$ function[a] | 12 and 16-19 December |
| VerBC_RT | Y | Modified according to observations on 11 December 2016 | 12 and 16-19 December |

[a] The values of $hs$ in VerBC_hs1, VerBC_hs2, VerBC_hs3, VerBC_hs4, VerBC_hs5 and
VerBC_hs6 are 0.35, 0.48, 0.53, 0.79, 0.82 and 0.96, respectively.



**Table 3.** Statistical metrics for temperature at 2 m (T2; °C), relative humidity at 2 m
(RH2; %), wind speed at 10 m (WS10; m s$^{-1}$), wind direction at 10 m (WD10, °), PM$_{2.5}$
(μg m$^{-3}$), SO$_2$ (ppbv), NO$_2$ (ppbv), CO (ppmv) and O$_3$ (ppbv).

| Variables | SIM[a] | OBS[b] | R[c] | MB[d] | NMB[e] | MFB[f] |
|---|---|---|---|---|---|---|
| T2 (°C) | -0.5 | -0.6 | 0.77 | 0.1 | -17.8 | -13.1 |
| RH2 (%) | 52.5 | 55.8 | 0.75 | -3.4 | -6.0 | -0.3 |
| WS10 (m s$^{-1}$) | 1.8 | 2.3 | 0.52 | -0.5 | -20.6 | -11.5 |
| WD10 (°) | 165.6 | 182.0 | 0.45 | -16.4 | -9.0 | 0.7 |
| PBLH (m) | 205.8 | 174.9 | 0.72 | 30.9 | 17.7 | 72.9 |
| PM$_{2.5}$ (μg m$^{-3}$) | 145.6 | 132.3 | 0.77 | 13.2 | 10.0 | 15.7 |
| SO$_2$ (ppbv) | 7.9 | 7.8 | 0.38 | 0.1 | 0.0 | -2.9 |
| NO$_2$ (ppbv) | 47.7 | 39.2 | 0.78 | 8.5 | 21.6 | 20.2 |
| CO (ppmv) | 1.8 | 1.9 | 0.73 | -0.1 | -4.9 | 6.4 |
| O$_3$ (ppbv) | 6.7 | 6.8 | 0.66 | -0.1 | -1.2 | -36.0 |

[a,b] SIM and OBS represent the averaged model results and observations in Beijing from 11 to 19
December 2016.
[c] R is the correlation coefficient which is calculated between hourly observations and simulations
in Beijing from 11 to 19 December 2016, $R = \frac{\sum_{i=1}^{n}|(OBS_i - OBS) * (SIM_i - SIM)|}{\sqrt{\sum_{i=1}^{n}(OBS_i - OBS)^2 + \sum_{i=1}^{n}(SIM_i - SIM)^2}}$, where OBS$_i$ and
SIM$_i$ are the hourly observed and simulated data in Beijing and n is the total number of hours.
[d] MB is the mean bias, $MB = \frac{1}{n} * \sum_{i=1}^{n} SIM_i - OBS_i$.
[e] NMB is the normalized mean bias, $NMB = \frac{1}{n} * \sum_{i=1}^{n} \frac{SIM_i - OBS_i}{OBS_i} * 100\%$.
[f] MFB is the mean fraction bias, $MFB = \frac{2}{n} * \sum_{i=1}^{n} \frac{SIM_i - OBS_i}{SIM_i + OBS_i} * 100\%$ .





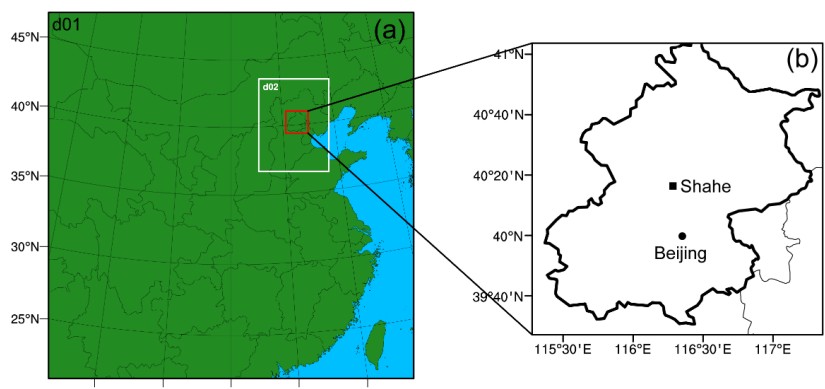

**Figure 1.** (a) Two nested domains with grid resolutions of 30 km (d01) and 10 km (d02).

(b) The BC vertical profiles were modified for the red box which covers the whole of

Beijing.



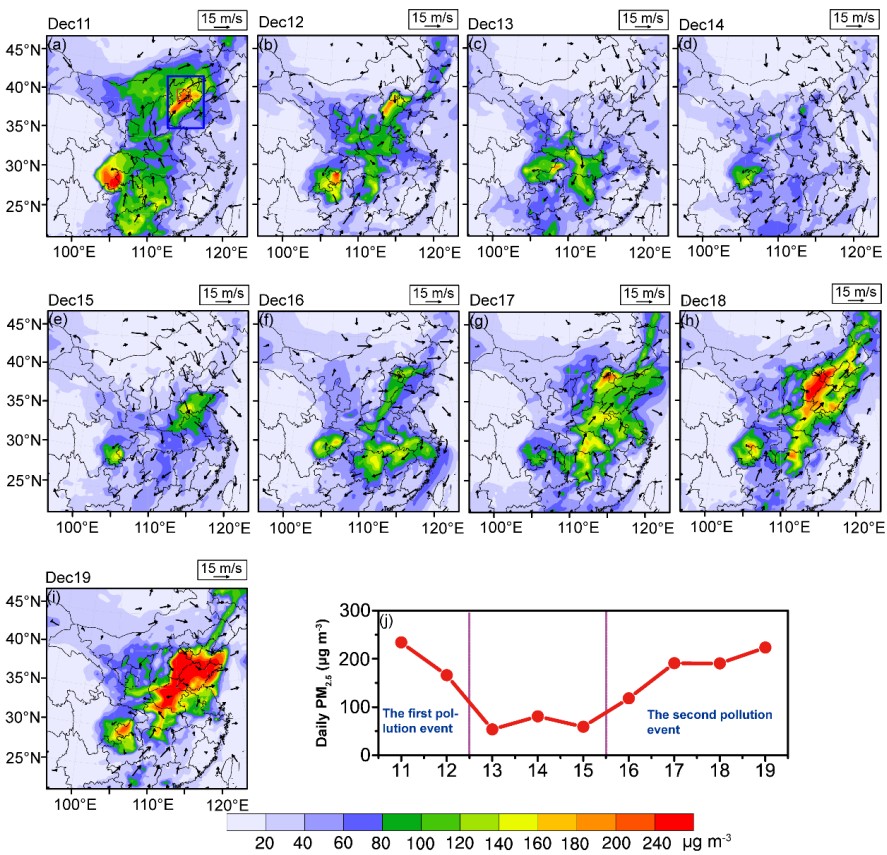

**Figure 2.** (a-i) Simulated spatial distributions of PM$_{2.5}$ concentrations (μg m$^{-3}$) and winds (m s$^{-1}$) at 850 hPa at 2 pm local time from 11 to 19 December 2016. (j) Time series of simulated daily PM$_{2.5}$ concentration in Beijing from 11 to 19 December 2016. Blue and red squares in the first panel denote the Beijing-Tianjin-Hebei and Beijing region, respectively.





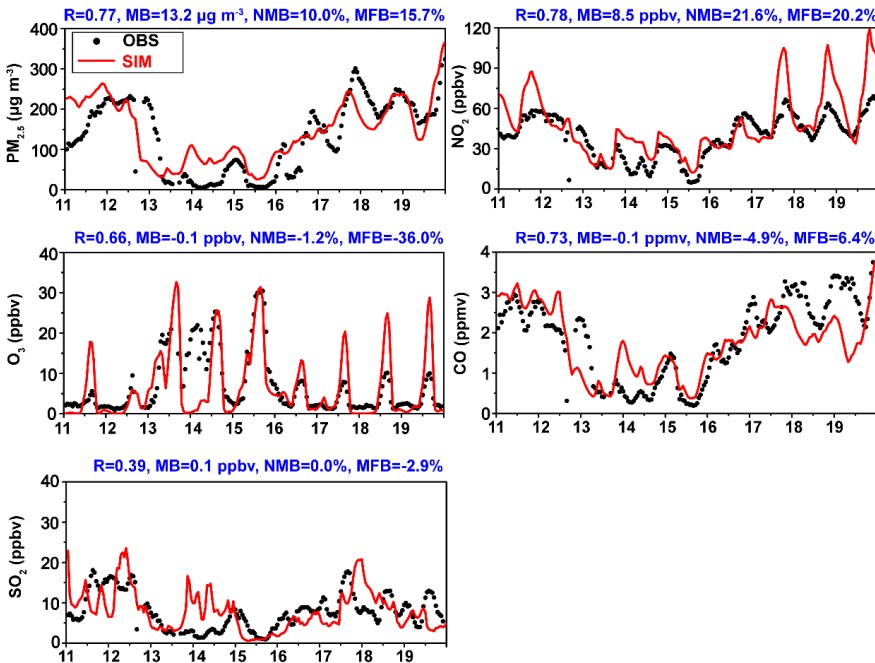

**Figure 3.** Time series of the observed (black dots) and simulated (red lines) hourly concentrations of $PM_{2.5}$ ($\mu g\ m^{-3}$), $NO_2$ (ppbv), $O_3$ (ppbv), CO (ppmv), $SO_2$ (ppbv) and BC ($\mu g\ m^{-3}$) in Beijing from 11 to 19 December 2016.

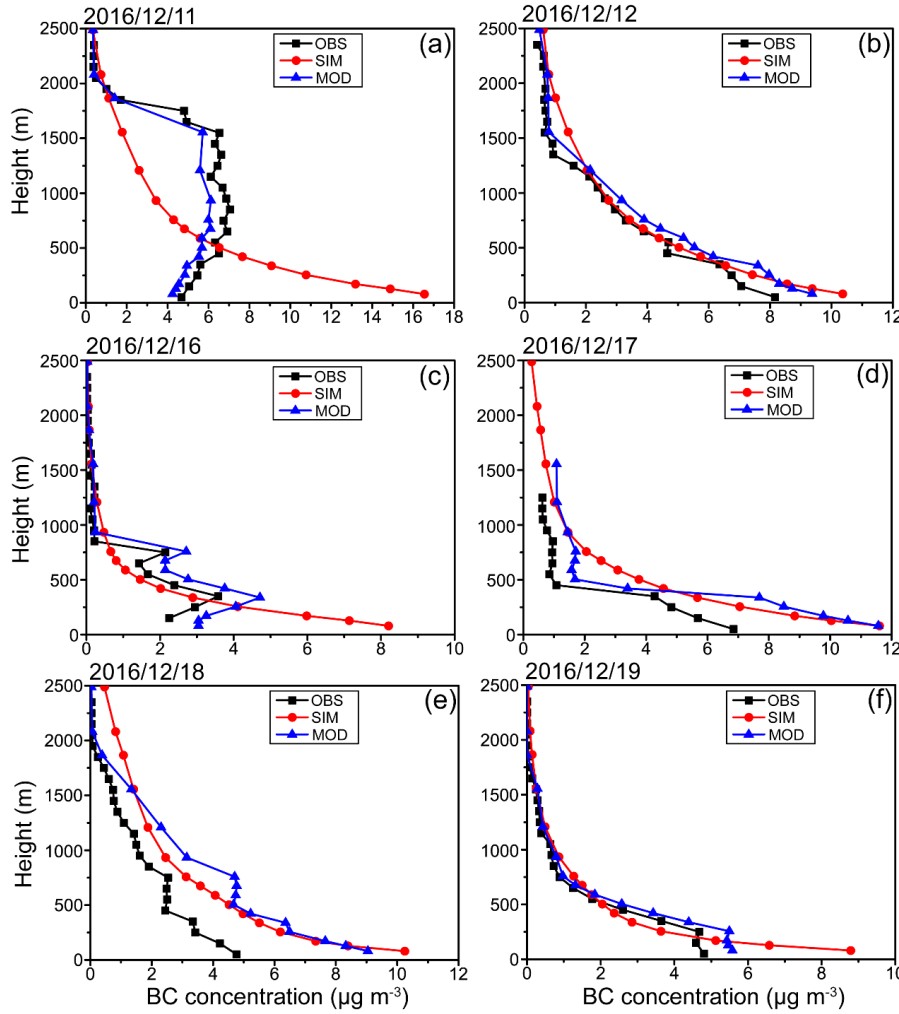

**Figure 4.** Observed (black line), simulated (red line) and modified (blue line) BC
vertical profiles in Beijing on 11-12 and 16-19 December 2016.

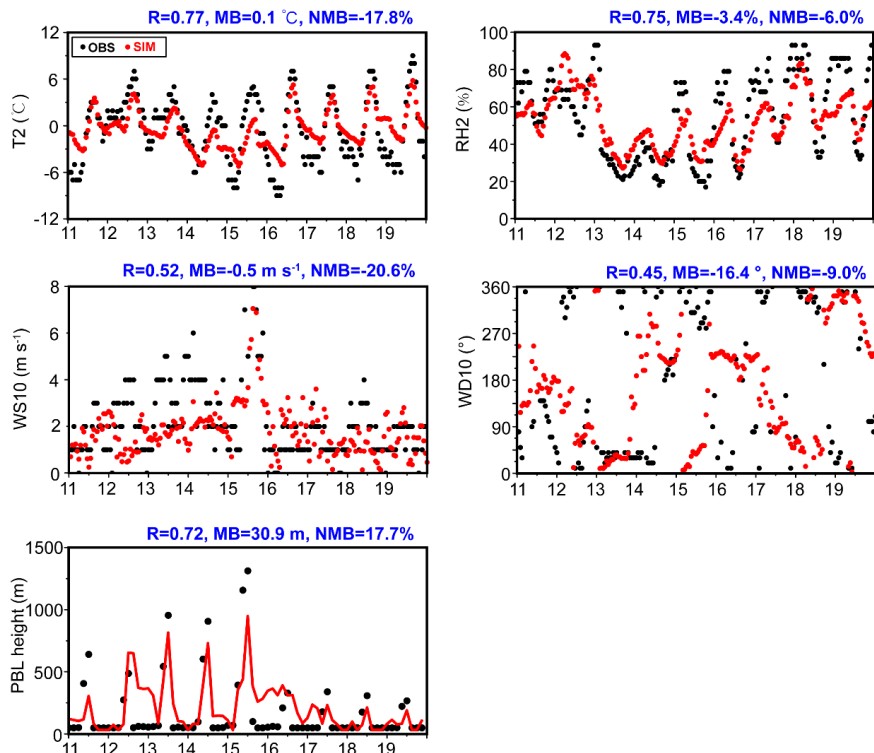


**Figure 5.** The black and red dots are the observed and simulated data of T2 (°C), RH2
(%), PBL height (m), WS10 (m s⁻¹) and WD10 (°) in Beijing from 11 December 2016
to 19 December 2016.





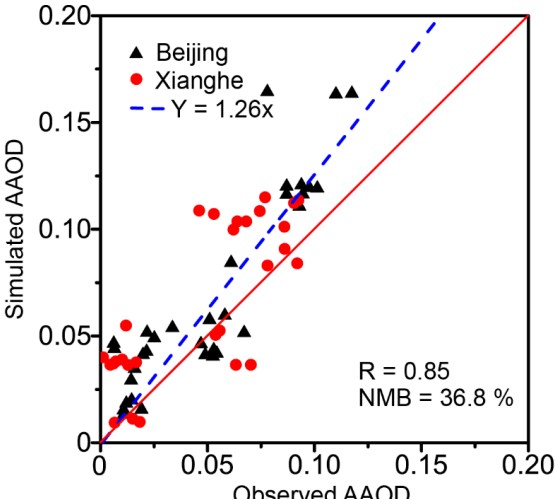

**Figure 6.** Comparison of simulated absorption aerosol optical depth (AAOD) at 550 nm with observations in Beijing (116.38°E, 39.98°N) and Xianghe (116.96°E, 39.75°N) station from 11 to 19 December 2016.

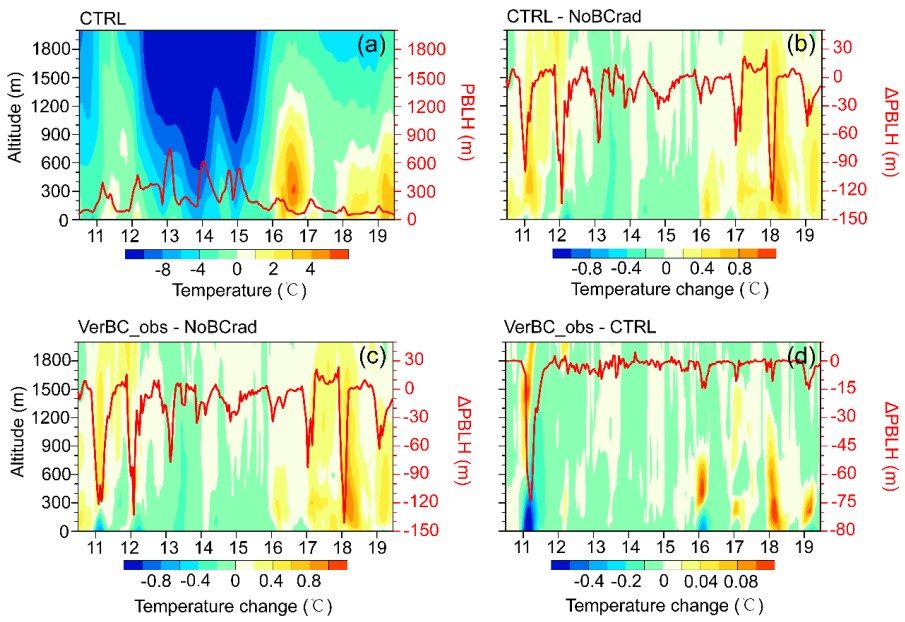

899

**Figure 7.** (a) Simulated hourly vertical profiles of temperature (contour) and PBLH

(red solid line) over Beijing at local time (LT) from 11 December 2016 to 19 December

2016. (b-d) Time series of changes in vertical temperature (contour) and PBLH

(ΔPBLH; red solid line) induced by BC DRE with original (b; CTRL minus NoBCrad)

and modified vertical profiles (c; VerBC_obs minus NoBCrad), and the difference

between the effects of modified and original BC profiles (d; VerBC_obs minus CTRL)

over Beijing region from 11 December 2016 to 19 December 2016.

907



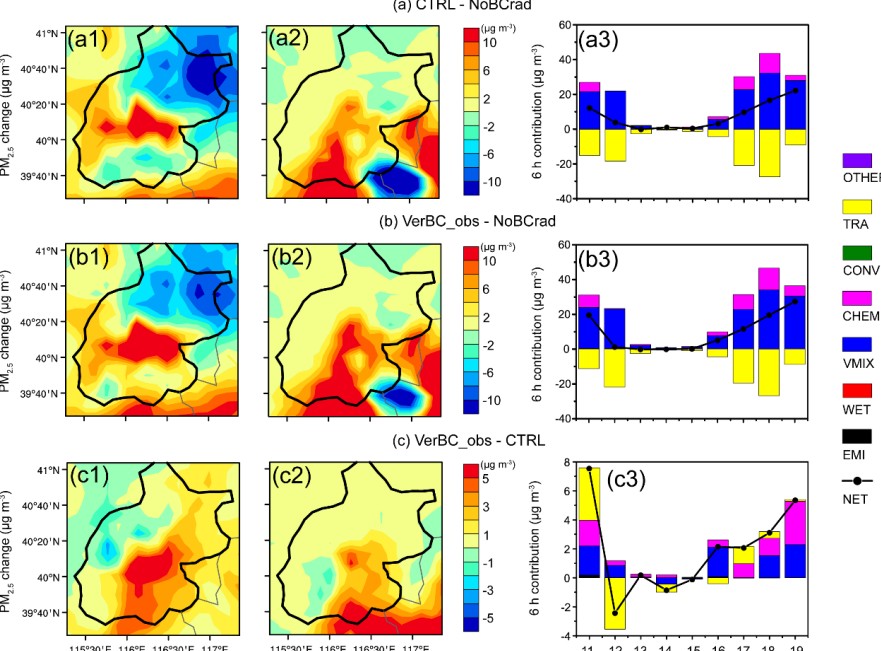

**Figure 8.** The spatial distribution of changes in near-surface $PM_{2.5}$ concentrations induced by BC DRE with original (CTRL minus NoBCrad; a1 and a2) and modified vertical profiles (VerBC_obs minus NoBCrad; b1 and b2), and the difference between VerBC_obs and CTRL (VerBC_obs minus CTRL; c1 and c2) over Beijing averaged over the period of 12:00 – 18:00 LT of the two haze events. a1-c1 represent the first pollution event of 11-12 December 2016 and a2-c2 represent the second pollution event of 16-19 December 2016. (a3-c3) The daily 6-h contributions of each physical/chemical process (colored bars, each of which is calculated as the concentration at 18:00 minus that at 12:00) to the change in $PM_{2.5}$ in Beijing from 11 December 2016 to 19 December 2016. The black dotted line represents the 6-h net contribution to $PM_{2.5}$ change by summing over all processes.



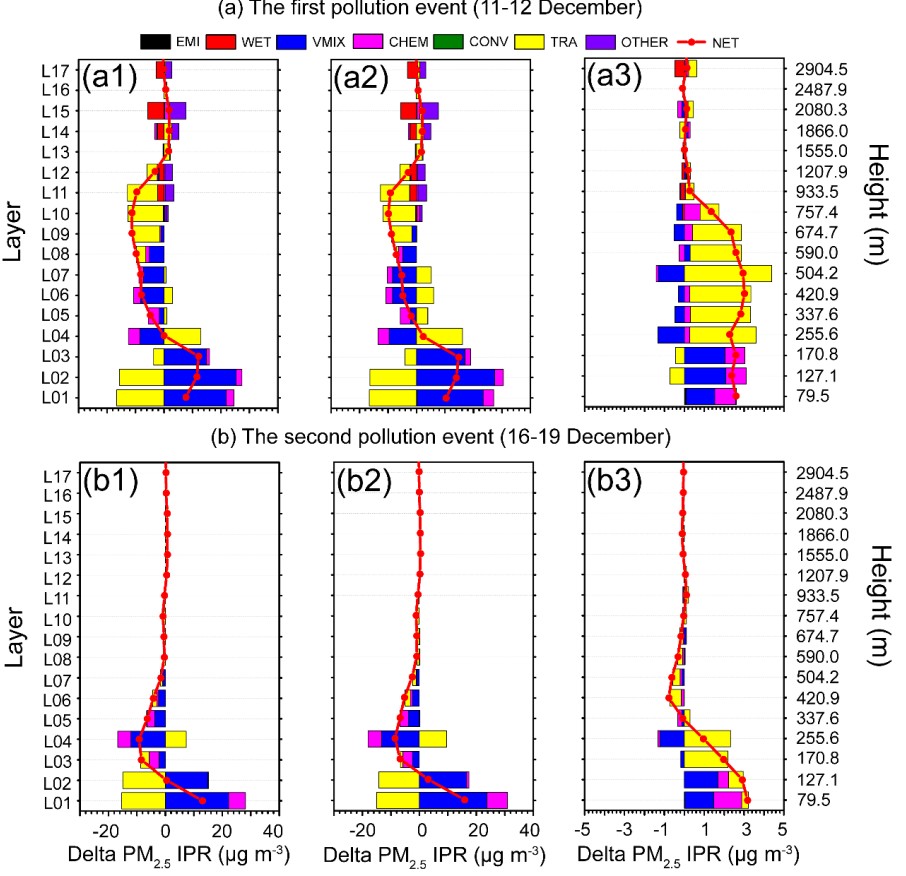

**Figure 9.** Vertical profiles of the 6-h contributions of physical/chemical processes (colored bars; each is calculated as the concentration at 18:00 LT minus that at 12:00 LT) to the changes in $PM_{2.5}$ induced by BC DRE with original (CTRL minus NoBCrad; a1 and b1) and modified vertical profiles (VerBC_obs minus NoBCrad; a2 and b2), and the difference between original and modified BC profiles (VerBC_obs minus CTRL; a3 and b3) over Beijing. The red dotted lines represent the 6-h net contributions to $PM_{2.5}$ changes by summing over all processes.

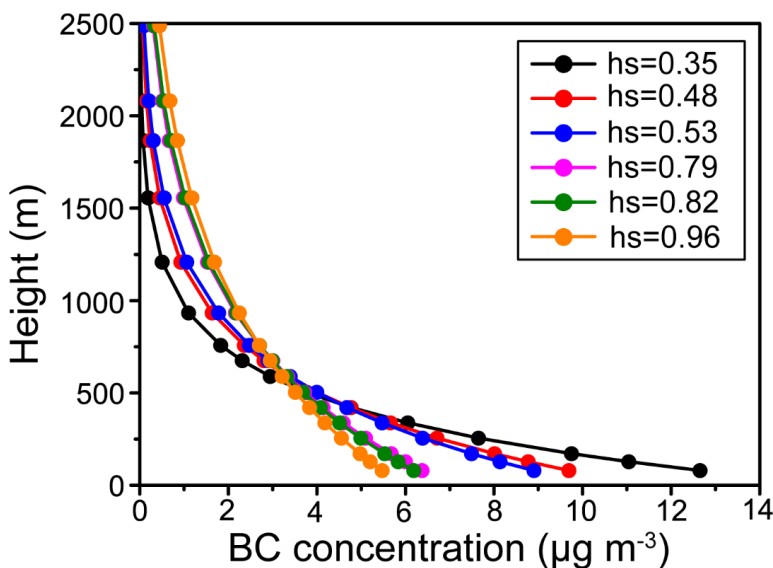

930

**Figure 10.** Vertical profiles of BC concentrations parameterized as six exponential

functions for 12 and 16-19 December 2016.

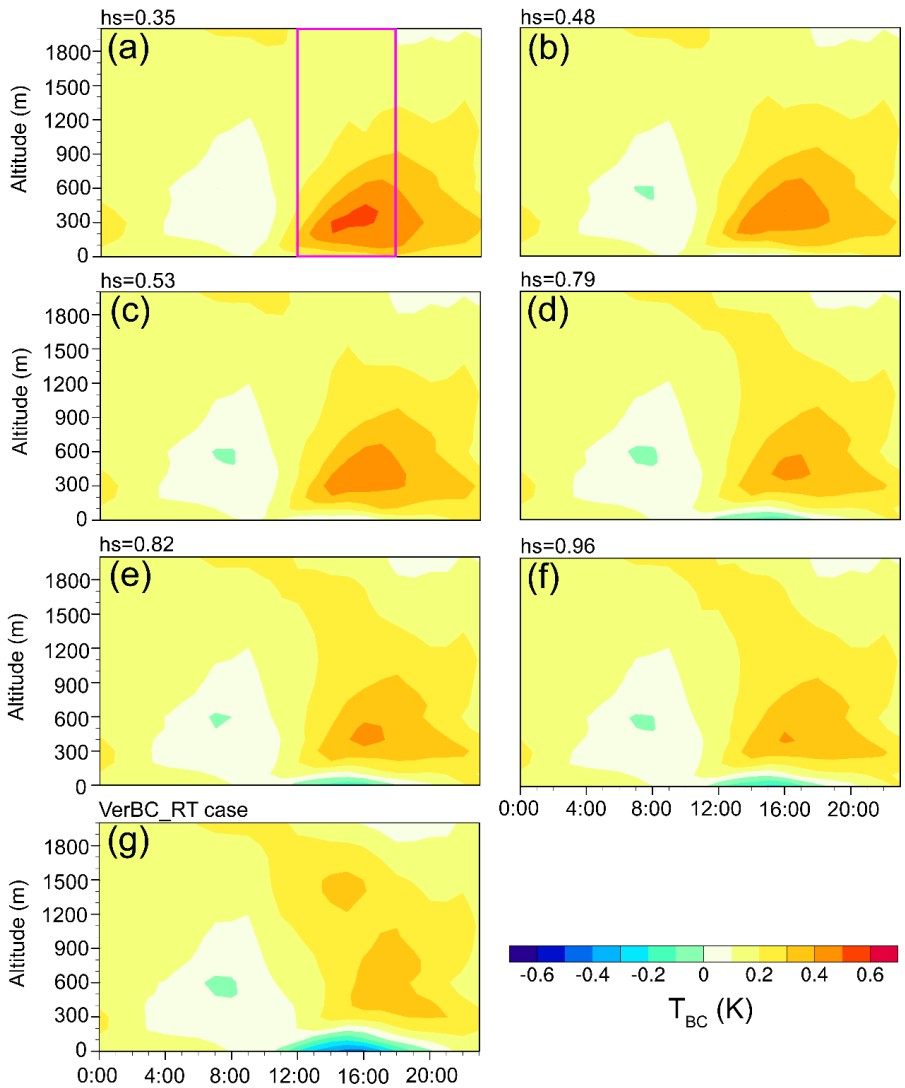

**Figure 11.** Time series of changes in vertical temperature induced by BC DRE with six exponential functions (VerBC_hs1-6 minus NoBCrad) and one transport-dominated vertical profile (VerBC_RT minus NoBCrad) averaged over 12 and 16-19 December 2016.





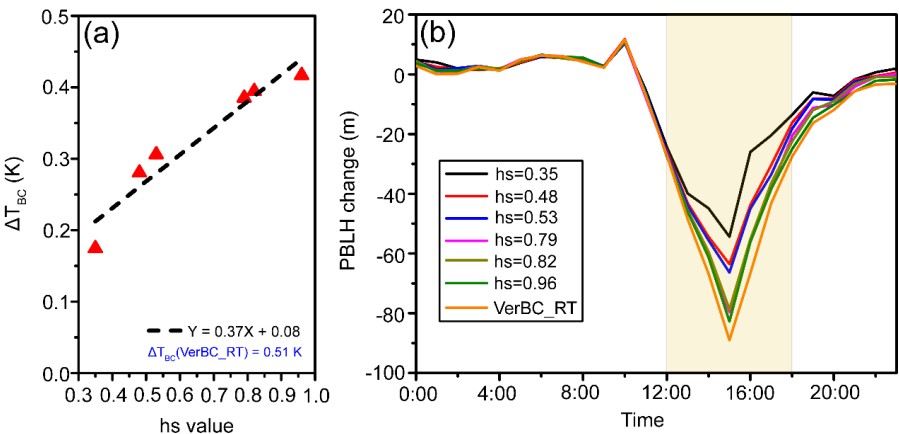

939

**Figure 12.** (a) Variation of $\Delta T_{BC}$ caused by BC DRE with increasing *hs* values averaged
12 and 16-19 December. The black dash line is the linear fit. (b) Time series of changes
in PBLH in Beijing caused by different BC vertical profiles averaged 12 and 16-19
December 2016.

944



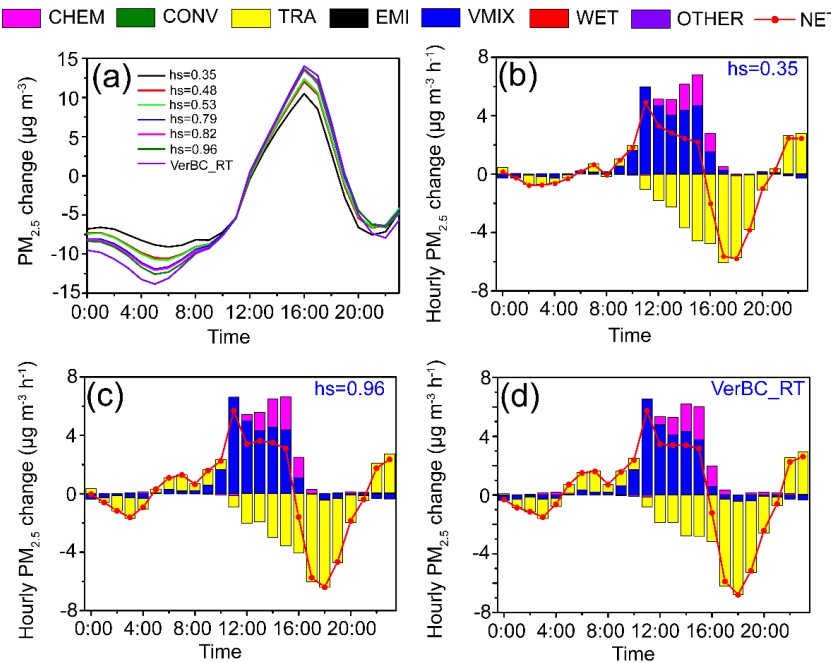

**Figure 13.** (a) Time series of the changes in surface-layer PM$_{2.5}$ in Beijing caused by BC with six exponential functions (VerBC_hs1-6 minus NoBCrad) and one observed transport-dominated vertical profile (VerBC_RT minus NoBCrad) averaged 12 and 16-19 December 2016. (b-d) The hourly contributions of each physical/chemical process to PM$_{2.5}$ changes caused by BC DRE with two exponential functions (*hs*=0.35 and 0.96) and one transport-dominated vertical profile.

