# Peer review of "Simulated impacts of vertical distributions of black carbon aerosol on meteorology and PM$_{2.5}$ concentrations in Beijing during severe haze events"

_Atmospheric Chemistry and Physics, 2021_

## Referee Comment (RC1)

**Review of "Simulated impacts of vertical distributions of black carbon aerosol on meteorology and PM2.5 concentrations in Beijing during severe haze events" by D. Chen et al.**

This study presents the impacts of vertical distributions of black carbon aerosol on local PM2.5 concentration and meteorology using the Weather Research and Forecasting with Chemistry model (WRF-Chem) and airborne measurement of black carbon vertical profiles. The manuscript is well written and easy to follow, and the simulated impacts are well documented and reported quantitatively. However, my main concern is that the scientific wisdom gained from this research, contributing/adding to the current knowledge of the community, is not clearly conveyed in the current form of the manuscript. In other words, I found multiple places where the authors can discuss more on the implications of the reported impacts/results, as well as the physical reasonings behind them, instead of only reporting the changes from one simulation to another. Therefore, I suggest major revision.

Overall, I think this is a nicely designed and conducted modelling study, which can make valuable contribution to the field. Here I provided some specific comments and suggested changes regarding my main concern of the manuscript.

**Major concerns/questions:**

Abstract:
A general comment, the current form reads like a report summary, could the authors reconstruct the abstract in a way that scientific questions/goals of the study are clearly posed in the beginning, followed by a concise summary of the key findings (not only reporting the quantitative statistics, but also the logical flows behind these changes), and ended with implications of the study.

Section 2.4 Numerical experiments:
When VerBC_obs and VerBC_hs1-6 (RT) are compared with noBCrad (which is ran with the default BC profiles, except the optical properties are set to zero), it seems you're attributing the simulated differences between them solely to radiative effects, while assuming difference in vertical BC profiles between these simulations and noBCrad has no non-radiative effects (e.g. microphysical or chemical effects). Could you please justify this?

Section 3 Model evaluation:
L309-311: Indeed the model overestimate PM2.5 concentration if one compares the averages over the 9 day period, however, there is a lot more one can say about this model/obs comparison. For example, it seems the overall overestimation in the simulation is mostly coming from the clean days (DEC 13-15), whereas during the 2 haze events, the model seems to do fairly good job, comparing to obs, quantitatively, but there seems to be a timing difference, which could be due to discrepancies in advection between obs & model. Only reporting the mean biases doesn't help the reader understand the difference between model and obs that much.

Moreover, what is the meteorological conditions during the clean period, is there any precipitation event? Cloud formation? These can also help the reader understand these events better. I wonder

if a meteorological overview of this 9-day period can be added to the beginning of Section 3 or section 3.2?

L350-353: Again, more details are needed here. Even though model overestimate PBLH in the mean, daily maximum PBLH values from obs exceed that from the model, and the overestimation is mainly due to the fact that obs has '~0m' PBLH during most of the day, is this an artifact or obs mis-characterize PBL? More details here would be helpful.

Section 4:
Is there attempt to compare the simulations with obs-corrected/modified BC vertical distribution to observation? I wonder if getting the vertical structure of BC close to obs help improve the overall simulation relative to observations? And this could be an important result of this study, such that getting observationally constrained BC vertical distribution help (or does not help) improve the simulated local meteorology and PM2.5 concentration.

L405-411: I feel like Figure S4 is an important figure which can be moved to the main text, as it shows the role of BCrad on regional circulation/wind pattern, which further leads to changes in local PM2.5 concentration. However, how does BC DRE enhance the northerlies north of NCP and weakened the wind speed in central and southern Beijing is still not clear to me. I also think investigating the physical mechanisms behind this is critical to the whole study. I suggest more detailed discussion and further analyses here.

Related to Fig. 9: What are the implications of these results from the IPR analyses?

Section 5:
L500-509: Could you discuss more on how does larger delta_T_BC result in larger reduction in PBLH?

Related to Fig. 13: Again, what are the implications of these results? Are these results case/event dependent? Or they can be generalized, e.g. to make arguments on the role of BC vertical distribution on local accumulation and advection of pollutant?

Conclusions:
Again, the current conclusion section reads like a summary of the simulation results, which feels redundant and repetitive. I suggest reconstruction of this section, in a way that the repetitive results summary can be minimized or summarized in higher level languages, and the authors are encouraged to discuss more on the implications of the results and the study in general, e.g. whether these results are event specific or they can be generalized. Languages on the role of BC vertical distribution in affecting local meteorology and transport/accumulation of pollutant should be added.

L580-582: You mentioned results from this study highlights the importance of accurate representation of BC vertical profiles in models. I don't think this point has been made clear, as you haven't shown how simulations with obs-modified BC profiles compared to observations, is there improvement at all?

Minor comments:

L44, L477: is "sharper" the right word? To me, the sharpest decline is associated with hs=0.35 where BC drops from 13 to 4 in the first 500m, whereas when hs=0.96, BC drops from 6 to 4 (much slower decline).

Section 2.1 Model configuration:
How does the current model configuration deal with properties at domain top, are the top of the domain forced by free-tropospheric properties/motions from NCEP?
Is there nudging in the simulation?

Section 2.3 Observational data:
Regarding MODIS dataset, did you use Aqua or Terra, at what resolution (i.e. what level of the dataset). More details here could be useful.

L275-280: A bit hard to follow here, on L275, you meant 'VerBC_hs1~6' instead of 'VerBC_hs1', right? When I first read this part, it confused me, but when I saw figure 11, it started to make sense. I suggest some clarifications here.

L367: I don't think "well simulate" is the right word to describe Fig. S3, even the horizontal distribution seems off between MODIS and simulation.

L414-418: repeated in the figure caption (Fig. 8), suggest reconstruction.

L498-499: not following this sentence, suggest rewording.

Figure 2: is the PM2.5 concentration shown here for near-surface or 850 hPa? Blue and red squares are hard to see on a printed copy. 2(i), could you also show BTH time series as well?

Figure 3: SO2 panel, is the NMB really 0.0% or a precision/rounding issue?

Figure 5: Wind speed and direction are critical to understanding the difference between obs and model, I wonder if daily or 6-hr (or even smaller intervals) averages of wind speed and direction can be shown to get a cleaner comparison between obs and model. It's pretty hard to compare the two in the current form of Fig. 5, especially WD10.

Figure 6: suggest indicating the 1:1 line with a different color (other than red, as it thought it was a fitting line for the red dots at first).

Figures 3, 5, 7, 8, 11: needs x-label titles, e.g. day, time …

---

## Referee Comment (RC2)

General comments:

In the current work, the authors used WRF-Chem with IPR analysis and sensitivity tests to assess the effect of vertical distribution of BC on meteorology and surface $PM_{2.5}$ concentrations during Beijing haze episodes. The research topic is very well suited to the scope of the current journal and the results presented are very interesting to their readers. However, there are some uncertainties, especially in the methodology, that require some modification before it is accepted. Without a concrete explanation of the methodology, the reviewer is uncertain whether the model settings chosen by the author will actually answer the authors' research questions. See the general and specific comments listed below.

1. There are no presentations on clouds and precipitation throughout the manuscript. In the presence of clouds and precipitation, the radiation effect of aerosols is quite different from that of sunny days. Please show and compare measured and simulated clouds, precipitations, or solar radiation.

Specific comments:

*Abstract:*
2. The authors simply repeated "$PM_{2.5}$ concentration", but more specifically, "$PM_{2.5}$ surface air concentrations" ($PM_{2.5}$ can be aloft). Please define "$PM_{2.5}$ concentration" as "surface air concentrations of $PM_{2.5}$" when it is first appeared.

3. P. 8, Ln. 154, "NCEP": Please be more specific. For example, specify the datasets number.

4. P. 9, Ln. 168, "FINN": Please provide horizontal and temporal resolution.

5. P. 10, Ln. 189, "updrafts": The convection scheme also includes downdrafts and precipitation. Which parameterization did you use for convection? Please list it in Table 1. The subgrid-scale wet deposition is calculated in the convection model so they can be counted as CONV here, but they need to be count as WET (wet deposition).

6. P. 10, Ln. 192, "cloud": How do you separate cloud chemical formation from in-cloud scavenging? The formation of $PM_{2.5}$ due to cloud chemistry occurs only when the cloud and rain droplets are completely evaporated. On the other hand, $PM_{2.5}$ in the cloud and

rain droplets are not counted as $PM_{2.5}$ and are removed from the air (although they are not completely removed unless droplets reach the ground).

7. P. 10, Ln. 194, "WET represents the wet removal processes of aerosols": In-cloud and below-cloud? Again, how do you separate "cloud" in CHEM from in-cloud scavenging in WET here?

8. P. 10, Ln. 195, "OTHER": What are they? Dry deposition should be the one but any other processes?

9. P. 10, Ln. 202: "Beijing station", where is it and what does it belong to? Does the station belong to NOAA? Maybe not, but the data was obtained from NOAA's website.

10. P. 10, Ln. 205, PBL of GDAS: what is the horizontal resolution of GDAS? Even though the GDAS is the analysis (or one can call it observation), is their PBL also assimilated with observed PBL? If not, PBL of GDAS cannot be regarded as observation as you show in Fig. 5. If it is assimilated with observed PBL, please specify which observation data was assimilated.

11. P. 12, "indirect radiative effects": how? The authors used the Lin's scheme for cloud microphysics, which is a single moment scheme and thus cloud albedo and cloud lifetime effects are not considered. Is it intended simulation settings?

12. P. 15-16, discrepancy of vertical profiles on December 11: Are the observation and simulation average times same? The simulated profile appears to be at night or very stable during the day, but the observed profile looks only be during the day.

13. It is necessary to discuss the reason for the difference in vertical profile between the simulation and observation on December 11th. Judging from the profile, the simulated surface air concentration is four times the observed value, but the overestimation of the simulated surface $PM_{2.5}$ concentration is not so high (Fig. 3). The simulated night T2 of the day has a significant overestimation (+6 deg C). It is 0 deg C in the simulation and -6 deg C in the observation. Is it due to overprediction of simulated clouds to prevent radiative cooling at night?

Figures

14. Caption of Fig. 1, "The BC vertical profiles were modified for the red box which covers …" should be written in the main text.

15. Fig. 2, "blue and red squares" are hardly legible.

16. Fig. 2(j), Does "Beijing" mean spatial average of the blue square region? Or one grid of the center of Beijing region? Please specify. Throughout the manuscript, it is hard to get whether the authors indicate values of only one grid point, one observation site, or those of spatial average.

17. Fig. 3, "Beijing". Again, Beijing point or Beijing area? Both for simulation and observation.

18. Fig. 3: Even though the model did not consider SOA, the simulated $PM_{2.5}$ was in perfect agreement with what was observed. Is the SOA negligibly small compared to the POA during the observation period, or do the OM/OC ratio(s) assumed for OC emission in the simulation well represent those of SOA and POA in the BTH region? Specify the number of OM/OC ratio(s) used in the simulation and how the author determined the value(s).

19. Caption of Fig. 4: What time? Both for observation and simulation.

20. Fig. 5: Can you compare downward solar radiation at ground surface here? It could also effectively evaluate the model performance of aerosols, and even clouds.

---

## Author Comment (AC1)

**Response to Comments of Reviewer #1**

**Manuscript number: acp-2021-611**

**Title:** Simulated impacts of vertical distributions of black carbon aerosol on meteorology and PM2.5 concentrations in Beijing during severe haze events

**General comments:**

This study presents the impacts of vertical distributions of black carbon aerosol on local  $PM_{2.5}$  concentration and meteorology using the Weather Research and Forecasting with Chemistry model (WRF-Chem) and airborne measurement of black carbon vertical profiles. The manuscript is well written and easy to follow, and the simulated impacts are well documented and reported quantitatively. However, my main concern is that the scientific wisdom gained from this research, contributing/adding to the current knowledge of the community, is not clearly conveyed in the current form of the manuscript. In other words, I found multiple places where the authors can discuss more on the implications of the reported impacts/results, as well as the physical reasonings behind them, instead of only reporting the changes from one simulation to another. Therefore, I suggest major revision.

Overall, I think this is a nicely designed and conducted modelling study, which can make valuable contribution to the field. Here I provided some specific comments and suggested changes regarding my main concern of the manuscript.

Thanks to the referee for the helpful comments and suggestions. We have revised the manuscript carefully and the point to point responses are listed below.

**Major concerns/questions:**

1. Abstract:

A general comment, the current form reads like a report summary, could the authors reconstruct the abstract in a way that scientific questions/goals of the study are clearly posed in the beginning, followed by a concise summary of the key findings (not only reporting the quantitative statistics, but also the logical flows behind these changes), and ended with implications of the study.

**Response:**

We have revised the abstract following the Reviewer's suggestion in revised manuscript.

2. Numerical experiments:

When VerBC\_obs and VerBC\_hs1-6 (RT) are compared with noBCrad (which is ran with the default BC profiles, except the optical properties are set to zero), it seems you're attributing the simulated differences between them solely to radiative effects, while assuming difference in vertical BC profiles between these simulations and noBCrad has no non-radiative effects (e.g. microphysical or chemical effects). Could you please justify this?

**Response:**

The experimental design in our study serves two purposes: (1) to compare the direct radiative effects (DRE) of BC with original and modified vertical profiles in two severe haze events and (2) to investigate the roles of parameterized BC vertical profiles in influencing meteorological conditions and  $PM_{2.5}$ . All the numerical experiments are summarized in Table 2 (see below).

For the first purpose, the differences in model results between CTRL (VerBC\_obs) and NoBCrad experiments (CTRL (VerBC\_obs) minus NoBCrad) represents the DRE of BC with original (modified) profiles on meteorology and PM2.5 concentrations.

For the second purpose, six BC profiles parameterized as exponential functions (VerBC\_hs1-6) and one profile of transport-dominated feature (VerBC\_RT) were considered, and the differences between VerBC\_hs1-6 and NoBCrad (VerBC\_hs1-6 minus NoBCrad) as well as the difference between VerBC\_RT and NoBCrad (VerBC\_RT minus NoBCrad) were quantified.

We have explained in the last paragraph of Section 2.4 that 'In VerBC\_obs, VerBC\_hs1-6, and VerBC\_RT experiments, the modifications of BC vertical profiles were performed only when the direct radiative effect of BC was calculated. All other physical and chemical processes in these experiments still used the original BC vertical profiles simulated by the model.'.

We have also explained in the second paragraph of Section 2.4 that 'In the case of NoBCrad, the BC DRE was turned off by setting the BC mass concentration equal to zero when calculating the optical properties of BC, following the studies of Qiu et al. (2017) and Chen et al. (2021).' Therefore the differences in BC profiles between VerBC obs (hs1-6, RT) and NoBCrad had no non-radiative effects in this study.

We have discussed the limitation of this study in the last paragraph of Section 6: 'There are channels for further improvement in near-future research. We distribute BC mass vertically according to the observed fractions of BC in individual model layers for each day without considering the hourly variations of BC vertical profiles due to the lack of data. Such assumed distribution of BC based on observation may not be consistent with the dynamical (winds, temperature, etc.) and chemical processes of the atmosphere. Further efforts are needed to examine the roles of BC vertical profiles in coupled chemistry-weather models.'.

|             | BC direct radiative effect (DRE) |                                             |                 |  |  |  |  |
|-------------|----------------------------------|---------------------------------------------|-----------------|--|--|--|--|
| Simulations | DRE                              | BC vertical profiles for calculation of DRE |                 |  |  |  |  |
|             | Turn on/off                      | Types description                           | Modified dates  |  |  |  |  |
| CTRL        | Y                                | Simulated by model                          | No modification |  |  |  |  |
| NoBCrad N   |                                  | Simulated by model                          | No modification |  |  |  |  |
| V-DC -1-    | V                                | Modified according to intraday              | 11-12 and 16-19 |  |  |  |  |
| verBC_obs   | Y                                | observations                                | December        |  |  |  |  |

| Table 2  | . Numerica     | l experiments.  | Y | indicates | "on". | and N | indicates | "off" |
|----------|----------------|-----------------|---|-----------|-------|-------|-----------|-------|
| I abit L | • I tufficited | in experiments. |   | marcates  | on,   |       | marcates  |       |

| VerBC_hs1-6 | V | Modified according to $C(h)=C_0$         | 12 and 16-19    |  |
|-------------|---|------------------------------------------|-----------------|--|
|             | 1 | $\times e^{-h/hs}$ function a | December        |  |
|             |   | Modified according to                    | 12  and  16  10 |  |
| VerBC_RT    | Y | observations on 11 December              | 12 and 10-19    |  |
|             |   | 2016                                     | December        |  |

a The values of *hs* in VerBC\_hs1, VerBC\_hs2, VerBC\_hs3, VerBC\_hs4, VerBC\_hs5 and VerBC hs6 are 0.35, 0.48, 0.53, 0.79, 0.82 and 0.96, respectively.

**3. Model evaluation:**

L309-311: Indeed the model overestimate  $PM_{2.5}$  concentration if one compares the averages over the 9 day period, however, there is a lot more one can say about this model/obs comparison. For example, it seems the overall overestimation in the simulation is mostly coming from the clean days (DEC 13-15), whereas during the 2 haze events, the model seems to do fairly good job, comparing to obs, quantitatively, but there seems to be a timing difference, which could be due to discrepancies in advection between obs & model. Only reporting the mean biases doesn't help the reader understand the difference between model and obs that much.

**Response:**

Following the Reviewer's suggestion, we have divided the studied period (11-19 December 2016) into: (1) the first pollution event (11-12 December), (2) the clean days (13-15 December), (3) the second pollution event (16-19 December) and have added the statistical metrics for PM2.5, SO2, NO2, CO and O3 for clean days and the two haze events as Table S2 (see below) of the Supplementary Material.

We have also added the following sentences to describe Table S2 in the second paragraph of Section 3.1: 'It should be noted that the model performance in simulating PM2.5, SO2, CO and O3 is better during the two haze events than on clean days. For hourly PM2.5, for example, the MBs (NMBs) are 29.1  $\mu$ g m-3 (82.5%) on clean days and 6.3  $\mu$ g m-3 (3.5%) during the two haze events. The possible reasons for the overall overestimation of PM2.5 are as follows: (1) the model biases in underestimating WS10 and daytime PBLH; (2) the uncertainties in anthropogenic emission data (e.g. the overestimation in the BC emissions) (Qiu et al., 2017; Chen et al., 2021). Overall, the model can capture fairly good the two severe pollution events in Beijing during 11-19 December 2016.'.

**Table S2.** Statistical metrics for PM2.5, SO2, NO2, CO and O3 on clean days and in two haze events.

| Variables                               | SIM                                                                                                                   | OBS                                                                                                                                      | R                                                                                                                                                               | MB                                                                                                                                                                                     | NMB                                                                                                                                                                                                         | MFB                                                                                                                                                                                                                                      |
|-----------------------------------------|-----------------------------------------------------------------------------------------------------------------------|------------------------------------------------------------------------------------------------------------------------------------------|-----------------------------------------------------------------------------------------------------------------------------------------------------------------|----------------------------------------------------------------------------------------------------------------------------------------------------------------------------------------|-------------------------------------------------------------------------------------------------------------------------------------------------------------------------------------------------------------|------------------------------------------------------------------------------------------------------------------------------------------------------------------------------------------------------------------------------------------|
| PM 2.5 (µg m -3 ) | 64.4                                                                                                                  | 35.3                                                                                                                                     | 0.15                                                                                                                                                            | 29.1                                                                                                                                                                                   | 82.5%                                                                                                                                                                                                       | 84.1%                                                                                                                                                                                                                                    |
| SO 2 (ppbv)                  | 5.5                                                                                                                   | 3.6                                                                                                                                      | -0.02                                                                                                                                                           | 1.9                                                                                                                                                                                    | 53.4%                                                                                                                                                                                                       | 18.8%                                                                                                                                                                                                                                    |
| NO 2 (ppbv)                  | 28.8                                                                                                                  | 20.8                                                                                                                                     | 0.55                                                                                                                                                            | 7.9                                                                                                                                                                                    | 38.0%                                                                                                                                                                                                       | 38.5%                                                                                                                                                                                                                                    |
| CO (ppmv)                               | 11.0                                                                                                                  | 14.2                                                                                                                                     | 0.64                                                                                                                                                            | -3.1                                                                                                                                                                                   | -22.0%                                                                                                                                                                                                      | -50.7%                                                                                                                                                                                                                                   |
|                                         | Variables
PM 2.5 (µg m -3 )
SO 2 (ppbv)
NO 2 (ppbv)
CO (ppmv) | Variables SIM   PM 2.5 (μg m -3 ) 64.4   SO 2 (ppbv) 5.5   NO 2 (ppbv) 28.8   CO (ppmv) 11.0 | Variables SIM OBS   PM 2.5 (μg m -3 ) 64.4 35.3   SO 2 (ppbv) 5.5 3.6   NO 2 (ppbv) 28.8 20.8   CO (ppmv) 11.0 14.2 | Variables SIM OBS R   PM 2.5 (μg m -3 ) 64.4 35.3 0.15   SO 2 (ppbv) 5.5 3.6 -0.02   NO 2 (ppbv) 28.8 20.8 0.55   CO (ppmv) 11.0 14.2 0.64 | Variables SIM OBS R MB   PM 2.5 (μg m -3 ) 64.4 35.3 0.15 29.1   SO 2 (ppbv) 5.5 3.6 -0.02 1.9   NO 2 (ppbv) 28.8 20.8 0.55 7.9   CO (ppmv) 11.0 14.2 0.64 -3.1 | Variables SIM OBS R MB NMB   PM 2.5 (μg m -3 ) 64.4 35.3 0.15 29.1 82.5%   SO 2 (ppbv) 5.5 3.6 -0.02 1.9 53.4%   NO 2 (ppbv) 28.8 20.8 0.55 7.9 38.0%   CO (ppmv) 11.0 14.2 0.64 -3.1 -22.0% |

|        | O 3 (ppbv)                   | 0.9   | 0.7   | 0.18 | 0.2  | 30.0% | 37.6%  |
|--------|-----------------------------------------|-------|-------|------|------|-------|--------|
|        | PM 2.5 (µg m -3 ) | 186.1 | 179.8 | 0.64 | 6.3  | 3.5%  | 8.0%   |
| Two    | SO 2 (ppbv)                  | 9.1   | 9.9   | 0.29 | -0.7 | -7.4% | -13.5% |
| haze   | NO 2 (ppbv)                  | 57.2  | 48.2  | 0.70 | 8.9  | 18.5% | 12.5%  |
| events | CO (ppmv)                               | 4.6   | 3.2   | 0.88 | 1.4  | 43.0% | -39.4% |
|        | O 3 (ppbv)                   | 2.2   | 2.4   | 0.30 | -0.2 | -9.3% | -8.4%  |

Moreover, what is the meteorological conditions during the clean period, is there any precipitation event? Cloud formation? These can also help the reader understand these events better. I wonder if a meteorological overview of this 9-day period can be added to the beginning of Section 3 or section 3.2?

**Response:**

As suggested, we have added a meteorological overview of this 9-day period at the beginning of Section 3.2. The observed hourly precipitation (mm) and 3-hourly total cloud cover (%) in Beijing are added as Figures 5g-5h (see below).

'The first haze event started on December 11 when southeasterlies transported polluted air from southern BTH to Beijing (Fig. 2a). Although the southeasterlies turned into northeasterlies in Beijing on December 12,  $PM_{2.5}$  concentrations were still high because of the high relative humidity (63.2%) that was conductive to the formation of secondary aerosols. With the relatively high wind speed of 3.6 m s-1 and low relative humidity of 37.2% in Beijing during 13-15 December, the haze pollution gradually disappeared (Fig. 2c-2e). From 16 to 19 December,  $PM_{2.5}$  began to accumulate again with unfavorable diffusion conditions (WS10 of 1.4 m s-1) and enhanced formation of secondary aerosols under high relative humidity of 67.1% (Li et al., 2019; Dai et al., 2021). Throughout the simulated period of 11-19 December 2016, Beijing had no precipitation and was partly cloudy (Fig. 5g-5h).'.

We have also added the following sentences in the first paragraph of Section 3.2 to describe the Fig. 5f-5h: 'The simulated SWDOWN in CTRL experiment agrees well with the observations with R and MB of 0.76 and -14.9 W m-2. Due to the limitation of the model outputs, the model provides only information of whether there is cloud in the grid or not. The model can reproduce well the presence of cloud during 11-19 December 2016. Both observations and model results show no precipitation in the studied time period.'

---

## Author Comment (AC2)

**Response to Comments of Reviewer #2**

**Manuscript number: acp-2021-611**

**Title:** Simulated impacts of vertical distributions of black carbon aerosol on meteorology and PM2.5 concentrations in Beijing during severe haze events

**General comments:**

In the current work, the authors used WRF-Chem with IPR analysis and sensitivity tests to assess the effect of vertical distribution of BC on meteorology and surface  $PM_{2.5}$  concentrations during Beijing haze episodes. The research topic is very well suited to the scope of the current journal and the results presented are very interesting to their readers. However, there are some uncertainties, especially in the methodology, that require some modification before it is accepted. Without a concrete explanation of the methodology, the reviewer is uncertain whether the model settings chosen by the author will actually answer the authors' research questions. See the general and specific comments listed below.

Thanks to the referee for the helpful comments and suggestions. We have revised the manuscript carefully and the point to point responses are listed below.

**Major concerns/questions:**

1. There are no presentations on clouds and precipitation throughout the manuscript. In the presence of clouds and precipitation, the radiation effect of aerosols is quite different from that of sunny days. Please show and compare measured and simulated clouds, precipitations, or solar radiation.

**Response:**

The comparison between simulated and observed hourly precipitation (mm), 3-hourly total cloud cover (%) and 3-hourly shortwave downward radiation flux (SWDOWN, W m-2) in Beijing are added in Figures 5f-5h (see below). The corresponding statistical metrics are added in Table 3 (see below). We have also added the following sentences in the first paragraph of Section 3.2 to describe the comparison.

'The simulated SWDOWN in CTRL experiment agrees well with the observations with R of 0.76 and MB of  $-14.9 \text{ W m}^{-2}$ . Due to the limitation of the model outputs, the model provides only information of whether there is cloud in the grid or not. The model can reproduce well the presence of cloud during 11-19 December 2016. Both observations and model results show no precipitation in the studied time period.'.

| Table 3. Statistical metrics for temperature at 2 m (T2; °C), relative humidity at 2 m                                                                  |
|---------------------------------------------------------------------------------------------------------------------------------------------------------|
| (RH2; %), wind speed at 10 m (WS10; m s -1 ), wind direction at 10 m (WD10, °), PBLH                                                         |
| (m), SWDOWN (W m -2 ), PM 2.5 (µg m -3 ), SO 2 (ppbv), NO 2 (ppbv), CO (ppmv) and O 3 |
| (ppby).                                                                                                                                                 |

| (FF ) |                         |                  |      |        |                  |                             |
|--------------|-------------------------|------------------|------|--------|------------------|-----------------------------|
| Variables    | SIM a | OBS b | R°   | $MB^d$ | NMB e | $\mathbf{MFB}^{\mathrm{f}}$ |
| T2 (°C)      | -0.5                    | -0.6             | 0.77 | 0.1    | -17.8%           | -13.1%                      |
| RH2 (%)      | 52.5                    | 55.8             | 0.75 | -3.4   | -6.0%            | -0.3%                       |

| WS10 (m s -1 )               | 1.8   | 2.3   | 0.52 | -0.5  | -20.6% | -11.5% |
|-----------------------------------------|-------|-------|------|-------|--------|--------|
| WD10 (°)                                | 165.6 | 182.0 | 0.45 | -16.4 | -9.0%  | 0.7%   |
| PBLH (m)                                | 205.8 | 174.9 | 0.72 | 30.9  | 17.7%  | 72.9%  |
| SWDOWN (W m -2 )             | 86.0  | 100.8 | 0.76 | -14.9 | -14.8% | -17.4% |
| PM 2.5 (µg m -3 ) | 145.6 | 132.3 | 0.77 | 13.2  | 10.0%  | 15.7%  |
| SO 2 (ppbv)                  | 7.9   | 7.8   | 0.38 | 0.1   | 1.4%   | -2.9%  |
| NO 2 (ppbv)                  | 47.7  | 39.2  | 0.78 | 8.5   | 21.6%  | 20.2%  |
| CO (ppmv)                               | 1.8   | 1.9   | 0.73 | -0.1  | -4.9%  | 6.4%   |
| O 3 (ppbv)                   | 6.7   | 6.8   | 0.66 | -0.1  | -1.2%  | -36.0% |